# Diffeomorphic interpolation for efficient persistence-based topological optimization

**Mathieu Carrière**
DataShape
Centre Inria d'Université Côte d'Azur
Sophia Antipolis, France
mathieu.carriere@inria.fr

**Marc Theveneau**[*]
Shape Analysis Group
Computer Science department, McGill
Montréal, Quebec, Canada
marc.theveneau@mail.mcgill.ca

**Théo Lacombe**
Laboratoire d'Informatique Gaspard Monge,
Univ. Gustave Eiffel, CNRS, LIGM, F-77454
Marne-la-Vallée, France
theo.lacombe@univ-eiffel.fr

## Abstract

Topological Data Analysis (TDA) provides a pipeline to extract quantitative topological descriptors from structured objects. This enables the definition of topological loss functions, which assert to what extent a given object exhibits some topological properties. These losses can then be used to perform topological optimization via gradient descent routines. While theoretically sounded, topological optimization faces an important challenge: gradients tend to be extremely sparse, in the sense that the loss function typically depends on only very few coordinates of the input object, yielding dramatically slow optimization schemes in practice.

Focusing on the central case of topological optimization for point clouds, we propose in this work to overcome this limitation using *diffeomorphic interpolation*, turning sparse gradients into smooth vector fields defined on the whole space, with quantifiable Lipschitz constants. In particular, we show that our approach combines efficiently with subsampling techniques routinely used in TDA, as the diffeomorphism derived from the gradient computed on a subsample can be used to update the coordinates of the full input object, allowing us to perform topological optimization on point clouds at an unprecedented scale. Finally, we also showcase the relevance of our approach for black-box autoencoder (AE) regularization, where we aim at enforcing topological priors on the latent spaces associated to fixed, pre-trained, black-box AE models, and where we show that learning a diffeomorphic flow can be done once and then re-applied to new data in linear time (while vanilla topological optimization has to be re-run from scratch). Moreover, reverting the flow allows us to generate data by sampling the topologically-optimized latent space directly, yielding better interpretability of the model.

## 1 Introduction

Persistent homology (PH) is a central tool of Topological Data Analysis (TDA) that enables the extraction of quantitative topological information (such as, e.g., the number and sizes of loops, connected components, branches, cavities, etc) about structured objects (such as graphs, times series or

---

[*]Part of this work was done when MT was doing an internship at the Laboratoire d'Informatique Gaspard Monge and student at Université Paris-Saclay.

38th Conference on Neural Information Processing Systems (NeurIPS 2024).

point clouds sampled from, e.g., submanifolds), summarized in compact descriptors called *persistence diagrams* (PDs). PDs were initially used as features in Machine Learning (ML) pipelines; due to their strong invariance and stability properties, they have been proved to be powerful descriptors in the context of classification of time series [42, 19], graphs [5, 23, 24], images [2, 25, 16], shape registration [7, 6, 36], or analysis of neural networks [22, 3, 26], to name a few.

Another active line or research at the crossroad of TDA and ML is *(persistence-based) topological optimization*, where one wants to modify an object $X$ so that it satisfies some topological constraints as reflected in its persistence diagram $\mathrm{Dgm}(X)$. The first occurrence of this idea appears in [21], where one wants to deform a point cloud $X \in \mathbb{R}^{n \times d}$ so that $\mathrm{Dgm}(X)$ becomes as close as possible (w.r.t. an appropriate metric denoted by $W$) to some target diagram $D_{\mathrm{target}}$, hence yielding to the problem of minimizing $X \mapsto W(\mathrm{Dgm}(X), D_{\mathrm{target}})$. This idea has then been revisited with different flavors, for instance by adding topology-based terms in standard losses in order to regularize ML models [14, 33, 29], improving ML model reconstructions by explicitly accounting for topological features [16], or improving correspondences between 3D shapes by forcing matched regions to have similar topology [36]. Formally, this goes through the minimization of an objective function

$$L : X \mapsto \ell(\mathrm{Dgm}(X)) \in \mathbb{R},$$

where $\ell$ is a user-chosen loss function that quantifies to what extent $\mathrm{Dgm}(X)$ reflects some prescribed topological properties inferred from $X$. Under mild assumptions (see Section 2), the map $L$ is differentiable generically and its gradients are obtained as a byproduct of the computation of $\mathrm{Dgm}(X)$. However, these approaches are limited in practice by two major issues: $(i)$ the computation of $X \mapsto \mathrm{Dgm}(X)$ scales poorly with the size of $X$ (e.g., number of points $n$ in a point cloud, number of nodes in a graph, etc), and $(ii)$ the gradient $\nabla L(X)$ tends to be very *sparse*: if $X = (x_1, \ldots, x_n) \in \mathbb{R}^{n \times d}$ is a point cloud, $\nabla L(X)_i \neq 0$ for only very few indices $i \in \{1, \ldots, n\}$ (the corresponding points are called the *critical points* of the topological gradient, see Section 2.1).

**Related works.**    Several articles have studied topological optimization in the TDA literature. The standard, or *vanilla*, framework to define and study gradients obtained from topological losses was described in [4, 28], where the high sparsity and long computation times were first identified. To mitigate this issue, the authors of [34] introduced the notion of *critical set* that extends the usually sparse set of critical points in order to get a gradient-like object that would update more points in $X$. In [39], the authors used an average of the vanilla topological gradients of several subsamples to get a denser and faster gradient. On the theoretical side, the authors of [27] demonstrated that adapting the stratified structure induced by PDs to the gradient definition enables faster convergence.

**Limitations.**    Despite proposing interesting ways to accelerate gradient descent, the approaches mentioned above are still limited in the sense that their proposed gradients are not defined on the whole space, but only on a sparse subset of the current observation $X$, which still prevents their use in different contexts, that we investigate in our experiments (Section 4). First, when the data has more than tens of thousands of points, the number of subsamples needed to capture relevant topological structures (when using [39]), as well as the critical set computations (when using [34]), both become *practically infeasible*. Second, when optimizing the topology of datasets obtained as *latent spaces* of a *black-box autoencoder model* (i.e., an autoencoder with forbidden access to its architecture, parameters, and training), then $(a)$ the topological gradients of [39, 34] *cannot* be re-used to process new such datasets, and topological optimization has to be performed from scratch every time that new data comes in, $(b)$ this also impedes their *transferability*, as re-running gradient descent every time makes it very difficult to guarantee some stability for the final output, and finally $(c)$ one *cannot* generate new data by sampling the optimized latent spaces directly, as it would require to apply the sequence of reverted gradients (which are not well-defined everywhere).

**Contributions and Outline.**    In this article, we propose to replace the standard gradient $\nabla L(X)$ of (**??**) derived formally by a *diffeomorphism* $v : \mathbb{R}^d \to \mathbb{R}^d$ that *interpolates* $\nabla L(X)$ on its non-zero entries, that is $v(x_i) = \nabla L(X)_i$ for all $i \in I := \{j \mid \nabla L(X)_j \neq 0\}$ and is, in some sense, as smooth as possible. More precisely, our contribution is three-fold:

- We introduce a *diffeomorphic interpolation* of the vanilla topological gradient, which extends this gradient to a smooth and denser vector field defined on the whole space $\mathbb{R}^d$, and which is able to move a lot more points in $X$ at each iteration,

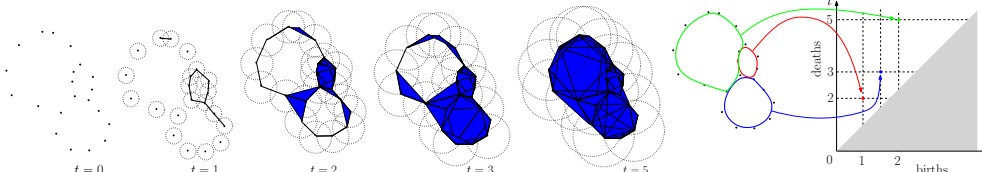

Figure 1: Illustration of the Vietoris-Rips filtration on a point cloud in $\mathbb{R}^d$, focusing on one-dimensional topological features (loops). When the filtration parameter $t$ increases, loops appear and disappear in the filtration. These values are accounted in the resulting persistence diagram (right).

- We prove that its updates indeed decrease topological losses, and we quantify its smoothness by upper bounding its Lipschitz constant (again, in the context of topological losses),
- We showcase its practical efficiency: we show that it compares favorably to the main baseline [34] in terms of convergence speed, that its combination with subsampling [39] allows to process datasets whose sizes are currently out of reach in TDA, and that it can successfully be used for the tasks mentioned above concerning black-box autoencoder models.

Section 2 provides necessary background in Topological Data Analysis and diffeomorphic interpolations. Section 3 presents our approach and its corresponding guarantees, and Section 4 showcases our experiments. Limitations and further research directions are discussed in Section 5.

## 2 Background

### 2.1 Topological Data Analysis

In this section, we recall the basic materials of Topological Data Analysis (TDA), and refer the interested reader to [20, 35] for a thorough overview. We restrict the presentation to our case of interest: extracting topological information from a point cloud using the standard *Vietoris-Rips* (VR) filtration. A more extensive presentation of the TDA machinery is provided in Appendix A.

Let $X = (x_1, \ldots, x_n) \in \mathbb{R}^{n \times d}$. The Vietoris-Rips filtration consists of building an increasing sequence of simplicial complexes $(K_t)_{t \geq 0}$ over $X$ by inserting a simplex $\sigma = (x_{i_1}, \ldots, x_{i_p})$ whenever $\forall j, j' \in \{1, \ldots, p\}$, $\|x_{i_j} - x_{i_{j'}}\| \leq t$. Each time a simplex $\sigma$ is inserted, it either creates a topological feature (e.g., inserting a face creates a cavity, that is a 2-dimensional topological feature) or destroy a pre-existing feature (e.g., the face insertion fills a loop, that is a 1-dimensional feature, making it topologically trivial). The persistent homology machinary tracks the apparition and destruction of such features in the so-called *persistence diagram* (PD) of $X$, denoted by $\mathrm{Dgm}(X)$. Therefore, $\mathrm{Dgm}(X)$ is a set of points in $\mathbb{R}^2$ of the form $(t_b, t_d)$ with $t_d \geq t_b$, where each such point accounts for the presence of a topological feature inferred from $X$ that appeared at time $t_b$ following the insertion of an edge $(x_{i_1}, x_{i_2})$ with $\|x_{i_1} - x_{i_2}\| = t_b$ and disappeared at time $t_d$ following the insertion of an edge $(x_{i_3}, x_{i_4})$ with $\|x_{i_3} - x_{i_4}\| = t_d$. Figure 1 illustrates this construction. From a computational standpoint, computing the VR diagram of $X \in \mathbb{R}^{n \times d}$ that would reflect topological features of dimension $d' \leq d$ runs in $O(n^{d'+2})$, making the computation quickly unpractical when $n$ increases, even when restricting to low-dimensional features such as connected components $(d' = 0)$, loops $(d' = 1)$ or cavities $(d' = 2)$.

**Topological optimization.** PDs are made to be used in downstream pipelines, either as static features (e.g., for classification purposes) or as intermediate representations of $X$ in optimization schemes. In this work, we focus on the second problem. We formally consider the minimization of objective functions of the form

$$L : X \in \mathbb{R}^{n \times d} \mapsto \ell(\mathrm{Dgm}(X)) \in \mathbb{R}. \tag{1}$$

Here, $\ell$ represents a loss function taking value from the space of PDs, denoted by $\mathcal{D}$ in the following. The space $\mathcal{D}$ can be equipped with a canonical metric, denoted by $W$ and whose formal definition is not required in this work, for which a central result is that the map $X \mapsto \mathrm{Dgm}(X)$ is stable (Lipschitz continuous) [17, 18, 38]. Therefore, if $\ell$ is Lipschitz continuous, so is $L$, hence it admits a gradient almost everywhere by Rademacher theorem. Building on these theoretical statements, one

can consider the "vanilla" gradient descent update $X_{k+1} := X_k - \lambda \nabla L(X_k)$ for a given learning-rate $\lambda > 0$ and iterate it in order to minimize (1). Theoretical properties of this seminal scheme (and natural extensions, e.g., stochastic gradient descent) have been studied in [4, 27], where convergence (to a local minimum of $L$) is proved.

From a computational perspective, deriving $\nabla L(X)$ comes in two steps. Let $\mu := \mathrm{Dgm}(X)$, written as $\mu = \{(b_i, d_i) \,|\, i \in I\}$ for some finite set of indices $I$. To each $i \in I$ correspond four (possibly coinciding) points $x_{i_1}, x_{i_2}, x_{i_3}, x_{i_4}$ in the input point cloud $X$. Intuitively, minimizing $\mu \mapsto \ell(\mu)$ boils down to prescribe a descent direction $(\delta b_i, \delta d_i) \in \mathbb{R}^2$ to each $(b_i, d_i)$ for $i \in I$, where $\delta b_i = \frac{\partial \ell}{\partial b_i}(\mu)$ and $\delta d_i = \frac{\partial \ell}{\partial d_i}(\mu)$. Backpropagating this perturbation to $X$ will move the corresponding points $x_{i_1}, x_{i_2}, x_{i_3}, x_{i_4}$ in order to increase or decrease the distances $\|x_{i_1} - x_{i_2}\| = b_i$ and $\|x_{i_3} - x_{i_4}\| = d_i$ accordingly. This yields to the following formula:

$$\frac{\partial L}{\partial x}(X) = \sum_{i, \; x \to (b_i, d_i)} \left[ \frac{\partial \ell}{\partial b_i} \cdot \frac{\partial b_i}{\partial x} + \frac{\partial \ell}{\partial d_i} \cdot \frac{\partial d_i}{\partial x} \right](X), \tag{2}$$

where the notation $x \to (b_i, d_i)$ means that $x \in X$ appears in (at least) one of the four points yielding the presence of $(b_i, d_i)$ in the diagram $\mu = \mathrm{Dgm}(X)$. A fundamental contribution of [28, §3.3] is to prove that the chain rule formula (2) is indeed valid[2]. Most of the time, a point $x \in X$ will not belong to any critical pair $(\sigma_b, \sigma_d)$ and the above gradient coordinate is 0. Therefore, the gradient of $L$ depends on very few points of $X$, yielding the sparsity phenomenon discussed in Section 1.

**Examples of common topological losses.** Let $X = (x_1, \ldots, x_n) \in \mathbb{R}^{n \times d}$ be a point cloud and $\mathrm{Dgm}(X) = \{(b_i, d_i) \,|\, i \in I\}$ be its PD. There are several natural losses that have been introduced in the TDA literature:

- Topological simplification losses: typically of the form $\sum_{i \in \tilde{I}}(b_i - d_i)^2$, where $\tilde{I} \subseteq I$. Such losses push (some of the) points in $\mathrm{Dgm}(X)$ toward the diagonal $\Delta = \{b = d\}$, hence destroying the corresponding topological features appearing in $X$.

- Topological augmentation losses [4]: similar to simplification losses, but typically attempting to push points in $\mathrm{Dgm}(X)$ away from $\Delta$, i.e., minimizing $-\sum_{i \in \tilde{I}}(b_i - d_i)^2$, to make topological features of $X$ more salient. As such losses are not coercive, they are usually coupled with regularization terms to prevent points in $X$ going to infinity.

- Topological registration losses [21]: given a target diagram $D_{\text{target}}$, one minimizes $W(\mathrm{Dgm}(X), D_{\text{target}})$ where $W$ denotes a standard metric between persistence diagrams. This loss attempts to modify $X$ so that it exhibits a prescribed topological structure.

## 2.2 Diffeomorphic interpolations

In order to overcome the sparsity of gradients appearing in TDA, we rely on diffeomorphic interpolations (see, e.g., [43, Chapter 8]). Let $X = (x_1, \ldots, x_n) \in \mathbb{R}^{n \times d}$, let $I \subseteq \{1, \ldots, n\}$ denote the set of indices on which $\nabla L(X)$ is non-zero and let $a_i := (\nabla L(X))_i \in \mathbb{R}^d$ for $i \in I$. Our goal is to find a smooth vector field $\tilde{v} : \mathbb{R}^d \to \mathbb{R}^d$ such that, for all $i \in I$, $\tilde{v}(x_i) = a_i$. To formalize this, we consider a Hilbert space $H \subset (\mathbb{R}^d)^{\mathbb{R}^d}$ for which the map $\delta_x^\alpha : f \mapsto \langle \alpha, f(x) \rangle_{\mathbb{R}^d} = \alpha^T f(x)$ is continuous for any $(\alpha, x) \in \mathbb{R}^d \times \mathbb{R}^d$. Such a space is called a Reproducing Kernel Hilbert Space (RKHS)[3]. A crucial property is that there exists a matrix-valued kernel operator $K : \mathbb{R}^d \times \mathbb{R}^d \to \mathbb{R}^{d \times d}$ whose outputs are symmetric and positive definite, and related to $H$ through the relation $\langle k_x^\alpha, k_y^\beta \rangle_H = \alpha^T K(x, y)\beta$ for all $x, y, \alpha, \beta \in \mathbb{R}^d$, where $k_x^\alpha \in H$ is the unique vector provided by the Riesz representation theorem such that $\langle k_x^\alpha, f \rangle = \langle \alpha, f(x) \rangle$. Conversely, any such kernel $K$ induces a (unique) RKHS $H$ (of which $K$ is the reproducing kernel). Now, we can consider the following problem:

$$\text{minimize } \|v\|_H, \text{ s.t. } v(x_i) = a_i, \; \forall i \in I, \tag{3}$$

that is, we are seeking for the smoothest (lowest norm) element of $H$ that solves our interpolation problem. The solution $\tilde{v}$ of this problem is the projection of 0 onto the affine set $\{v \in H \,|\, v(x_i) =$

---

[2]This is not trivial, because the intermediate space $\mathcal{D}$ is only a metric space.

[3]In many applications, RKHS are restricted to spaces of functions valued in $\mathbb{R}$ or $\mathbb{C}$, but the theory adapts faithfully to the more general setting of vector-valued maps.

$a_i, \forall i \in I\}$. Observe that $\tilde{v}$ belongs to the orthogonal of $\{v \in H \mid v(x_i) = 0, \forall i \in I\}$, and thus of $\{v \in H \mid \langle k_{x_i}^{\alpha_i}, v \rangle_H = 0, \ \forall i \in I, \alpha_i \in \mathbb{R}^d\}$, and therefore $\tilde{v} \in \mathrm{span}(\{k_{x_i}^{\alpha_i} \mid i \in I\})$. This justifies to search for $\tilde{v}$ in the form of $\tilde{v}(x) = \sum_{i \in I} K(x, x_i)\alpha_i$, and the interpolation that it must satisfy yields $\tilde{v}(x) = \sum_{i \in I} K(x, x_i)(\mathbb{K}^{-1}a)_i$, where $\mathbb{K}$ is the block matrix $(K(x_i, x_j))_{i,j \in I}$ and $a = (a_i)_{i \in I}$. See also [43, Theorem 8.8]. In particular, it is important to note that $\tilde{v}$ inherits from the regularity of $K$ and will typically be a diffeomorphism in this work. If $K$ is the Gaussian kernel defined by $K(x, y) := \exp\left(-\frac{\|x-y\|^2}{2\sigma^2}\right) I_d$ for some bandwidth $\sigma > 0$, a choice to which we stick to in the rest of this work, the expression of $\tilde{v}$ reduces to

$$\tilde{v}(x) = \sum_{i \in I} \rho_\sigma(\|x - x_i\|)\alpha_i, \tag{4}$$

where $\rho_\sigma(u) := e^{-\frac{u^2}{2\sigma^2}}$, and $\alpha_i := (\mathbf{K}^{-1}a)_i$ with $\mathbf{K} = (\rho_\sigma(\|x_i - x_j\|)I_d)_{i,j \in I}$. Note that $\tilde{v}$ can be understood as the convolution of $a$ with a Gaussian kernel, but involving a correction $\mathbf{K}^{-1}$ guaranteeing that after the convolution, the interpolation constraint is satisfied. We will call $\tilde{v}$ the *diffeomorphic interpolation* associated to the vectors and indices $\{a_i \mid i \in I\}$.

## 3 Diffeomorphic interpolation of the vanilla topological gradient

### 3.1 Methodology

We aim at minimizing a loss function $L : X \mapsto \ell(\mathrm{Dgm}(X))$ as in (1), starting from some initialization $X_0$, and assuming that $L$ is lower bounded (typically by 0) and locally semi-convex. This assumption is typically satisfied by the topological losses $\ell$ introduced in Section 2.1. Gradient descents implemented in practice are (explicit) discretization of the *gradient flow*

$$\frac{\mathrm{d}X}{\mathrm{d}t} \in -\partial L(X(t)), \quad X(0) = X_0, \tag{5}$$

where $\partial L(X) := \{v \mid L(Y) \geq L(X) + v \cdot (Y - X) + o(Y - X) \text{ for all } X, Y\}$ denotes the subdifferential of $L$ at $X$. Note that a topological loss $L$ is typically *not* differentiable everywhere, since the map $X \mapsto \mathrm{Dgm}(X)$ is differentiable almost everywhere but not in $C^{1,1}$. However, uniqueness of the gradient flow on a maximal interval $[0, +\infty[$ is guaranteed if $L$ is lower bounded and locally semi-convex [15, §B.1].

In this work, we propose to use the dynamic described by the diffeomorphism $\tilde{v}_t$ introduced in (4) interpolating the current vanilla topological gradient $\nabla L(X_t)$ at each time $t$, formally considering solutions $\tilde{X}$ of

$$\frac{\mathrm{d}\tilde{X}}{\mathrm{d}t} = -\tilde{v}_t(\tilde{X}(t)), \quad \tilde{X}(0) = X_0. \tag{6}$$

Here, slightly overloading notation, $\tilde{v}_t(\tilde{X}(t))$ denotes the $n \times d$ matrix where the $i$-th line is given by $\tilde{v}_t(\tilde{X}(t)_i)$. The *flow* at time $T$ associated to (6) is the map

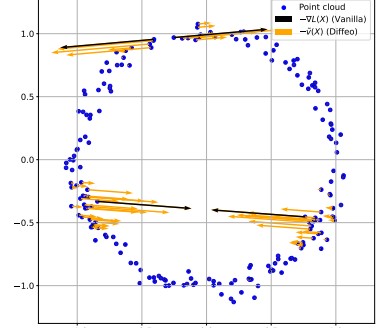

Figure 2: (blue) A point cloud $X$, and (black) the negative gradient $-\nabla L(X)$ of a simplification loss which aims at destroying the loop by collapsing the circle (reduce the loop's death time) and tearing it (increase the birth time). While $\nabla L(X)$ only affects four points in $X$, the diffeomorphic interpolation $\tilde{v}(X)$ (orange, $\sigma = 0.1$) is defined on $\mathbb{R}^d$, hence extends smoothly to other points in $X$.

$$\varphi_T : x_0 \mapsto x_0 - \int_0^T \tilde{v}_t(x(t))\mathrm{d}t, \quad \dot{x}(t) = -\tilde{v}_t(x(t)), \ x(0) = x_0, \tag{7}$$

which can inverted by simply following the flow backward (i.e., by following $\tilde{v}_t$ instead of $-\tilde{v}_t$). We now guarantee that at each time $t$, following $\tilde{v}_t$ instead of the vanilla topological gradient $\nabla L(X_t)$ still provides a descent direction for the topological loss $L$.

**Proposition 3.1.** *For each $t \geq 0$, it holds that $\frac{\mathrm{d}L(\tilde{X}(t))}{\mathrm{d}t} = -\|\nabla L(\tilde{X}(t))\|^2 \leq 0$.*

*Proof.* One has $\frac{\mathrm{d}L(\tilde{X}(t))}{\mathrm{d}t} = -\langle \nabla L(\tilde{X}(t)), \tilde{v}_t(\tilde{X}(t)) \rangle = -\sum_{i=1}^n (\nabla L(\tilde{X}(t)))_i \cdot (\tilde{v}_t(\tilde{X}(t)))_i$. Since $\nabla L(\tilde{X}(t))_i = 0$ for $i \notin I$, and $\tilde{v}_t(\tilde{X}(t))_i = -\nabla L(\tilde{X}(t))_i$ for $i \in I$, the result follows. $\square$

Moreover, it is also possible to upper bound the smoothness, i.e., the Lipschitz constant, of $\tilde{v}$:

**Proposition 3.2.** *Let $L$ be the simplification or augmentation loss computed with $k = |\tilde{I}|$ PD points, as defined at the end of Section 2.1. Let $\tilde{v} = \tilde{v}_t$ be the diffeomorphic interpolation associated to the vanilla topological gradient at time $t \geq 0$. Then, one has, $\forall x, y \in \mathbb{R}^d$ and $t \geq 0$:*

$$\|\tilde{v}(x) - \tilde{v}(y)\|_2 \leq \|\tilde{v}(x) - \tilde{v}(y)\|_1 \leq C_d \cdot \sigma^{d-1} \cdot \kappa(\mathbf{K}) \cdot \mathrm{Pers}_k(\mathrm{Dgm}(\tilde{X}(t))) \cdot \|x - y\|_2,$$

*where $C_d = \sqrt{d} \cdot 2^{3+\frac{d+1}{2}} \cdot \pi^{\frac{d-1}{2}}$, $\kappa(\mathbf{K})$ is the condition number of $\mathbf{K}$, and $\mathrm{Pers}_k(\mathrm{Dgm}(\tilde{X}(t)))$ is the sum of the $k$ largest distances to the diagonal in $\mathrm{Dgm}(\tilde{X}(t))$.*

The proof is deferred to Appendix B. This upper bound can be used to quantify how smooth the diffeomorphic interpolation is (as characterized with its Lipschitz constant) based on the parameters it is computed from. In the case of the Gaussian kernel, we found that our upper bound on the Lipschitz constant depends on the kernel bandwidth $\sigma$: indeed, the more spread the Gaussian function is, the more critical points can influence other data points potentially far from them, introducing unwanted distortions and larger Lipschitz constant. Similarly, if the condition number $\kappa(\mathbf{K})$ is large, inverting the kernel matrix might introduce instabilities in Equation (4), and thus a larger Lipschitz constant as well. Finally, the total persistence also appears, as the more PD points one has to optimize, the more critical pairs will appear, and thus the more constrained Equation (3) is, leading to more complex diffeomorphic interpolation solutions with larger Lipschitz constants.

## 3.2 Subsampling techniques to scale topological optimization

As a consequence of the limited scaling of the Vietoris-Rips filtration with respect to the number of points $n$ of the input point cloud $X$, it often happens in practical applications that computing the VR diagram $\mathrm{Dgm}(X)$ of a large point set $X$ (*a fortiori* its gradient) turns out to be intractable. A natural workaround is to randomly sample $s$-points from $X$, with $s \ll n$, yielding a smaller point cloud $X' \subset X$. Provided that the Hausdorff distance between $X'$ and $X$ is small, the stability theorem [18, 17, 8] ensures that $\mathrm{Dgm}(X')$ is close to $\mathrm{Dgm}(X)$. See [11, 12, 9] for an overview of subsampling methods in TDA.

However, the sparsity of vanilla topological gradients computed from topological losses strikes further when relying on subsampling: only a tiny fraction of the seminal point cloud $X$ is likely to be updated at each gradient step. In contrast, using the diffeomorphic interpolation $\tilde{v}$ (of the vanilla topological gradient) computed on the subsample $X'$ still provides a vector field defined on the whole input space $\mathbb{R}^d$, in particular on each point of $X$ and the update can then be performed in linear time with respect to $n$. This yields Algorithm 1. Figure 3 illustrates the qualitative benefits offered by the joint use of subsampling and diffeomorphic interpolations when compared to vanilla topological gradients. A larger-scale experiment is provided in Section 4.

---

**Algorithm 1** Diffeomorphic gradient descent for topological loss functions with subsampling

---

**Input:** Initial $X_0 \in \mathbb{R}^{n \times d}$, loss function $\ell$, learning rate $\lambda > 0$, subsampling size $s \in \{1, \ldots, n\}$, max. epoch $T \geq 1$, stopping criterion.
Set $L : X \mapsto \ell(\mathrm{Dgm}(X))$ (+ possibly a regularization term in $X$).
**for** $k = 1, \ldots, T$ **do**
    Subsample $X'_{k-1} = \{x'_1, \ldots, x'_s\}$ uniformly from $X_{k-1}$.
    Compute $\nabla L(X'_{k-1})$ (vanilla topological gradient)
    Compute the diffeomorphic interpolation $\tilde{v}(X'_{k-1})$ from $\nabla L(X'_{k-1})$ using (4).
    Set $X_k := X_{k-1} - \lambda \tilde{v}(X_{k-1})$.
    **if** stopping criterion is reached **then**
        **Return** $X_k$
    **end if**
**end for**
**Return** $X_T$

---

**Stopping criterion.** A natural stopping criterion for Algorithm 1 is to assess whether the loss $L(X_t) = \ell(\mathrm{Dgm}(X_t))$ is smaller than some $\varepsilon > 0$. However, computing $\mathrm{Dgm}(X_t)$ can be intractable if $X_t$ is large. Therefore, a tractable loss to consider is $\hat{L}(X_t) := \mathbb{E}[\ell(\mathrm{Dgm}(X'_t))]$, where

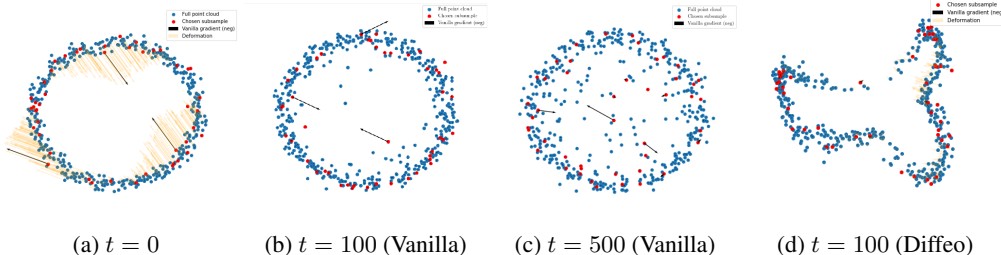

| (a) $t = 0$ | (b) $t = 100$ (Vanilla) | (c) $t = 500$ (Vanilla) | (d) $t = 100$ (Diffeo) |

Figure 3: Showcase of the usefulness of subsampling combined with diffeomorphic interpolations to minimize a topological simplification loss, with parameters $\lambda = 0.1$, $s = 50$, $n = 500$. ($a$) Initial point cloud $X$ (blue), subsample $X'$ (red), vanilla topological gradient on the subsample (black) and corresponding diffeomorphic interpolation (orange). ($b$) and ($c$), the point cloud $X_t$ after running $t = 100$ and $t = 500$ steps of vanilla gradient descent. ($d$) the point cloud $X_t$ after running $t = 100$ steps of diffeomorphic gradient descent.

$X'_t$ is a uniform $s$-sample from $X_t$. Under that perspective, Algorithm 1 can be re-interpreted as a kind of stochastic gradient descent on $\hat{L}$, for which two standard stopping criteria can be used: ($a$) compute an exponential moving average of the loss on individual samples $X'_t$ over iterations, or ($b$) compute a validation loss, i.e., sample $X'_{t,(1)}, \ldots, X'_{t,(K)}$ and estimate $\hat{L}$ by $K^{-1} \sum_{k=1}^{K} \ell(\mathrm{Dgm}(X'_{t,(k)}))$. Empirically, we observe that the latter approach with $K = n/s$ (more repetitions for smaller sample sizes to mitigate variance) yields the most satisfactory results (faster convergence toward a better objective $X_t$) overall, and thus stick to this choice in our experiments.

## 4  Numerical experiments

We provide numerical evidence for the strength of our diffeomorphic interpolations. PH-related computations relies on the library `Gudhi` [40] and automatic differentiation relies on `tensorflow` [1]. The "big-step gradient" baseline [34] implementation is based on `oineus`[4]. The first two experiments were run on a `11th Gen Intel(R) Core(TM) i5-1135G7 @ 2.40GHz`, the last one on a `2x Xeon SP Gold 5115 @ 2.40GHz`. Our code is publicly available at https://github.com/tlacombe/topt.

**Convergence speed and running times.** We sample uniformly $n = 200$ points on a unit circle in $\mathbb{R}^2$ with some additional Gaussian noise, and then minimize the simplification loss $L : X \mapsto \sum_{(b,d) \in \mathrm{Dgm}(X)} |d|^2$, which attempts to destroy the underlying topology in $X$ by reducing the death times of the loops by collapsing the points. The respective gradient descents are iterated over a maximum of 250 epochs, possibly interrupted before if a loss of 0 is reached ($\mathrm{Dgm}(X)$ is empty), with a same learning rate $\lambda = 0.1$. The bandwidth of the Gaussian kernel in (4) is set to $\sigma = 0.1$. We include the competitor `oineus` [34], as—even though relying on a fairly different construction—this method shares a key idea with ours: extending the vanilla gradient to move more points in $X$. We stress that both approaches can be used in complementarity: compute first the "big-step gradient" of [34] using `oineus`, and then extend it by diffeomorphic interpolation. Results are displayed in Figure 4[5]. In terms of loss decrease over *iterations*, both "big-step gradients" and our diffeomorphic interpolations significantly outperform vanilla topological gradients, and their combined use yields the fastest convergence (by a slight margin over our diffeomorphic interpolations alone). However, in terms of raw running times, the use of `oineus` involves a significant computational overhead, making our approach the fastest to reach convergence by a significant margin.

**Subsampling.** We now showcase how using our diffeomorphic interpolation jointly with subsampling routines (Algorithm 1) allows to perform topological optimization on point clouds with thousands of points, a new scale in the field. For this, we consider the vertices of the Stanford Bunny [41], yielding a point cloud $X_0 \in \mathbb{R}^{n \times d}$ with $n = 35,947$ and $d = 3$. We consider a topolog-

---

[4] https://github.com/anigmetov/oineus

[5] Note that the loss computed with `oineus` has *not* been normalized in the figure, which is why its values are larger than the others. This has no influence over its minimum number of iterations needed to reach 0 though.

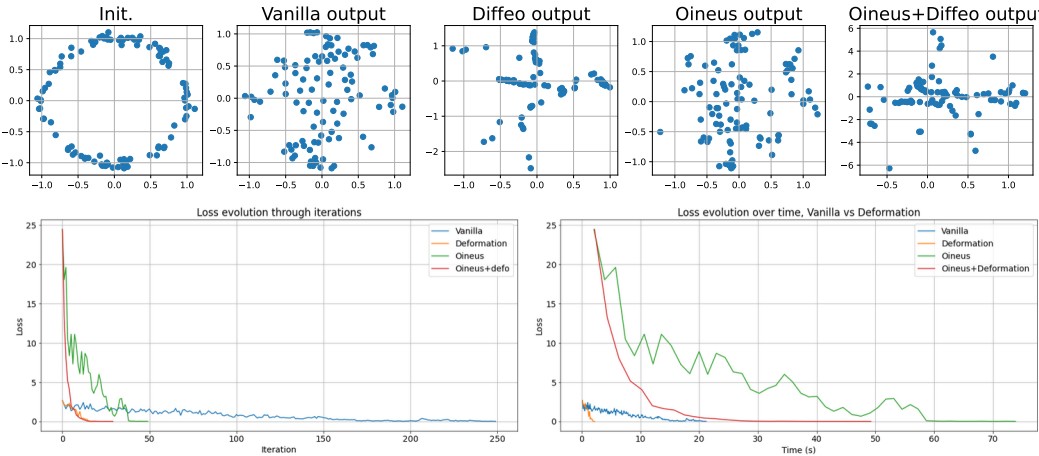

Figure 4: (Top) From left to right: initial point cloud, and final point cloud for the different flows. (Bottom) Evolution of the losses with respect to the number of iterations and with respect to running time.

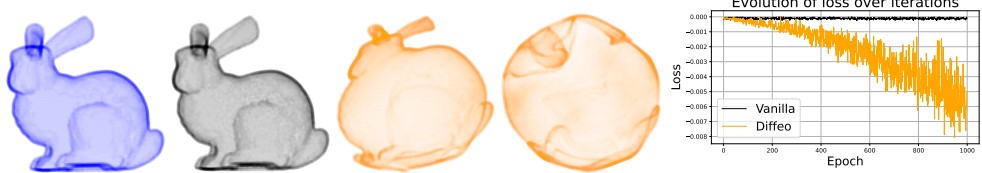

Figure 5: From left to right: initial Stanford bunny $X_0$, the point cloud after $1,000$ epochs of vanilla topological gradient descent (barely any changes), the point cloud after 200 epochs of diffeomorphic gradient descent, after 1,000 epochs, and eventually the evolution of losses for both methods over iterations.

ical augmentation loss (see Section 2.1) for two-dimensional topological features, i.e., we aim at increasing the persistence of the bunny's cavity. The size of $n$ makes the computation of $\mathrm{Dgm}(X_0)$ untractable (recall that it scales in $O(n^4)$); we thus rely on subsampling with sample size $s = 100$ and compare the vanilla gradient descent scheme and our scheme described in Algorithm 1. Results are displayed in Figure 5. Because it only updates a tiny fraction of the initial point cloud at each iteration, the vanilla topological gradient with subsampling barely changes the point cloud (nor decreases the loss) in 1,000 epochs. In sharp contrast, as our diffeomorphic interpolation computed on subsamples is defined on $\mathbb{R}^3$, it updates the whole point cloud at each iteration, making possible to decrease the objective function where the vanilla gradient descent is completely stuck. Note that a step of diffeomorphic interpolation, in that case, takes about 10 times longer than a vanilla step. An additional subsampling experiment can be found in Appendix C.

**Black-box autoencoder models.** Next, we apply our diffeomorphic interpolations to black-box autoencoder models. In their simplest formulation, autoencoders (AE) can be summarized as two maps $E : \mathbb{R}^d \to \mathbb{R}^{d'}$ and $D : \mathbb{R}^{d'} \to \mathbb{R}^d$ called encoder and decoder respectively. The intermediate space $\mathbb{R}^{d'}$ in which the encoder $E$ is valued is referred to as a *latent space* (LS), with typically $d' \ll d$. In general, without further care, there is no reason to expect that the LS of a point cloud $X$, $E(X) = \{E(x_1), \ldots, E(x_n)\}$, reflects any geometric or topological properties of $X$. While this can be mitigated by adding a topological regularization term to the loss function *during the training* of the autoencoder [29, 4], this *cannot* work in the setting where one is given a *black-box, pre-trained* AE. However, replacing $(E, D)$ by $(\varphi \circ E, D \circ \varphi^{-1})$ for any invertible map $\varphi : \mathbb{R}^{d'} \to \mathbb{R}^{d'}$ yields an AE producing the same outputs yet changing the LS $E(X)$, without explicit access to the AE's model. Hence, we propose to learn such a $\varphi$ with diffeomorphic interpolations: given some latent space $E(X)$, we apply $T$ steps of our diffeomorphic gradient descent algorithm to $X \mapsto \ell(\mathrm{Dgm}(X))$ initialized at $E(X)$. We thus get a sequence of smooth displacements $-\tilde{v}_1, \ldots, -\tilde{v}_T$ of $\mathbb{R}^{d'}$ that discretizes the flow (7) via $\varphi : x_0 \mapsto x_0 - \sum_{k=1}^{T} \tilde{v}_k(x_{k-1})$ where $x_k - x_{k-1} = -\tilde{v}_k(x_{k-1})$, and such that $\mathrm{Dgm}(\varphi(E(X)))$ is more topologically satisfying. This fixed diffeomorphism $\varphi$ can

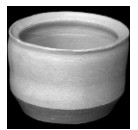 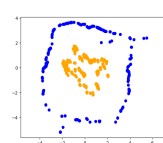 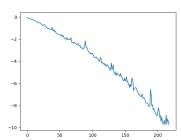 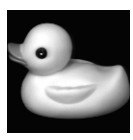 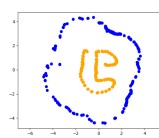 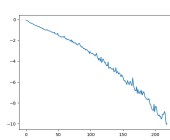

Figure 7: COIL images, their corresponding initial LSs in blue and final LSs obtained with diffeomorphic gradient descent in orange, and the corresponding topological losses, for both vase (left) and duck (right).

then be re-applied to any new data coming out of the encoder in a deterministic way. Moreover, any random sample from the topologically-optimized LS can be inverted without further computations by following $\tilde{v}_T, \tilde{v}_{T-1}, \ldots, \tilde{v}_1$, which allows to push the new sample back to the initial LS, and then apply the decoder on it. Again, this cannot be achieved with baselines [39, 34].

In order to illustrate these properties, we trained a variational autoencoder (VAE) to project a family of datasets of images representing rotating objects, named COIL [32], to two-dimensional latent spaces. Given that every dataset in this family is comprised of 288 pictures of the same object taken with different angles, one can impose a prior on the topology of the corresponding LSs, namely that they are sampled from circles. However, the VAE architecture is shallow (the encoder has one fully-connected layer (100 neurons), and the decoder has two (50 and 100 neurons), all layers use ReLu activations), and thus the learned latent spaces, although still looking like curves thanks to continuity, do not necessarily display circular patterns. This makes generating new data more difficult, as latent spaces are harder to interpret. To improve on this, we learn a flow $\varphi$ as described above with an augmentation loss associated to the 1-dimensional PD point which is the most far away from the diagonal, in order to force latent spaces to have a significant one-dimensional topological feature, i.e., loop. As the datasets are small, we do not use subsampling, and we use learning rate $\lambda = 0.1$, Gaussian kernels with bandwidth $\sigma = 0.3$ and an increase of at least 3. in the topological loss (from an iteration to the next) to stop the algorithm[6].

We provide some qualitative results in Figure 7 (see also Appendix C, Figure 10). In order to quantify the improvement, we also computed the Pearson correlation scores between the ground-truth angles $\theta_i$ and the angles $\hat{\theta}_i$ computed from the topologically-optimized LSs with

$$\hat{\theta}_i := \angle(\varphi \circ E(x_i) - \hat{\mathbb{E}}[\varphi \circ E(X)], \varphi \circ E(x_1) - \hat{\mathbb{E}}[\varphi \circ E(X)]),$$

where $\hat{\mathbb{E}}[\varphi \circ E(X)] := n^{-1} \sum_{i=1}^{n} \varphi \circ E(x_i)$, and $\varphi$ denotes our flow. In Table 1, we provide an average of these scores over 100 test sets obtained by randomly perturbing the training set with uniform noise of amplitude 0.05. As expected, correlation becomes better after forcing the latent spaces to have the topology of a circle. This better interpretability is also illustrated in Figure 6, in which four angles are specified, which are mapped to the topologically-optimized LS, then pushed to the initial LS of the black-box VAE by following the reverted flow of our learned diffeomorphism $\varphi$, and finally decoded back into realistic, COIL-like images.

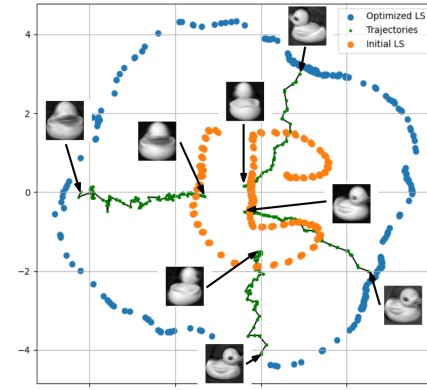

Figure 6: As the four samples from the topologically-optimized LS (blue) are far from the initial LS (orange), the decoded images are fuzzy. However, reverting $\varphi$ and following the corresponding green trajectories allows to render good-looking images.

**Single-cell data.** We also deployed our proposed topological correction of LSs from AEs on a real-world dataset of single cells. Specifically, we designed an experiment on single cell HiC (scHiC) data inspired from [27]. The dataset is comprised of single cells characterized by chromatin folding, that is, each cell is encoded by the spatial distance matrix of its DNA fragments. The dataset we focus on is taken from [31], in which it was shown that cells are sampled at different phases of the cell cycle. Thus, similar to COIL images, we expect latent embeddings of this dataset to exhibit a circular shape, that we can constrain with diffeomorphic topological optimization.

---

[6]Indeed, we noticed that 3. was a consistent threshold for detecting whether the representative cycle of the most persistent PD point changed between iterations.

| Dataset | Duck | Cat | Pig | Vase | Teapot |
|---|---|---|---|---|---|
| No optim. | $0.56 \pm$ 7.5e-04 | $0.78 \pm$ 1.4e-03 | $0.17 \pm$ 5.7e-04 | $0.86 \pm$ 7.2e-03 | $0.32 \pm$ 1.6e-03 |
| Diffeo | **0.61 $\pm$ 3.1e-03** | **0.83 $\pm$ 1.2e-03** | **0.76 $\pm$ 2.1e-04** | **0.93 $\pm$ 9.8e-04** | **0.39 $\pm$ 3.4e-03** |

| Dataset | scHiC (augmentation) | scHiC (registration) |
|---|---|---|
| No optim. | $0.79 \pm$ 8.1e-03 | $0.792 \pm$ 8.1e-03 |
| Diffeo | **0.84 $\pm$ 4.3e-03** | **0.794 $\pm$ 8.4e-03** |

Table 1: Means and variances of correlation scores computed over 100 test sets, for both `COIL` and scHiC.

Specifically, we processed this single cell dataset of $1,171$ cells with the stratum-adjusted correlation coefficient (SCC) with 500kb and convolution parameter $h = 1$ on chromosome 10. Then, we run kernel PCA on the SCC matrix to obtain a preprocessed dataset in $\mathbb{R}^{100}$, on which we applied the same VAE architecture than the one described above for `COIL` images. Finally, we optimized two losses, the first was the same augmentation loss than for the `COIL` images, the second was the following registration loss:

$$L : X \in \mathbb{R}^{n \times d} \mapsto W^2(\mathrm{Dgm}^1(X), D_{\mathrm{target}}),$$

where $\mathrm{Dgm}(X)$ contains the points of the PD of $X$ with distance-to-diagonal at least $\tau = 1$, $D_{\mathrm{target}}$ is a target PD with only one point $p^* = [-3.5, 3.5]$, and $W$ is the 2-Wasserstein distance between PDs. We used $\sigma = 0.2$ for the augmentation loss and $\sigma = 0.025$ for the registration loss ($\sigma$ is set to a smaller value for the registration loss in order to mitigate the effects of matching instability), $\lambda = 0.1$ on 500 epochs, with subsampling of size $s = 300$ for computational efficiency (as computing VR diagrams without radius thresholding on $1,171$ points already takes few minutes on a laptop CPU, which becomes hardly tractable if done repetitively as in gradient descent), and loss increase of 3. as a stopping criterion. Qualitative results are displayed in Appendix C, Figure 11, and we also measured quantitative improvement with the correlation scores between latent space angles and repli scores[7] in Table 1, in which improvements can be observed. An additional experiment on the influence of the bandwidth parameter $\sigma$ over these correlation scores, as well as over convergence, can also be found in Appendix C.

## 5 Conclusion

In this article, we have presented a way to turn sparse topological gradients into dense diffeomorphisms with quantifiable Lipschitz constants, and showcased practical benefits of this approach in terms of convergence speed, scaling, and applications to black-box AE models on several datasets. Several questions are still open for future work.

In terms of theoretical results, we plan on working on the stability between the diffeomorphic interpolations computed on a dataset and its subsamples. This requires some control over the locations of the critical points, which we expect to be possible in *statistical estimation*; indeed sublevel sets of density functions are know to have stable critical points [10, Lemma 17]. We also plan to look at *adaptive kernels* whose parameters (like the bandwidth $\sigma$) depend on the input point cloud $\sigma = \sigma(X)$ (instead of using a fixed kernel and parameters at every iteration), and understand the convergence properties of our proposed diffeomorphic gradient descent. Finally, applying our diffeomorphic interpolation to sparse gradients computed with *multiparameter persistent homology* is another natural research direction, provided that differentiability properties have recently been proved in that setting [30, 37].

Concerning the AE experiment, we plan to investigate the limitations presented in the figure below:
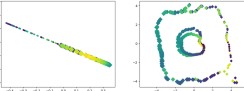 as the initial LSs (colored with the ground-truth angles) have zero (left) or two (right) loops, it is impossible to unfold them with diffeomorphisms; instead the optimized latent spaces either also exhibit no topology, or mixes different angles. In future work, we plan on investigating other losses or gradient descent schemes for diffeomorphic topological optimization, including stratified procedures similar to [27] that allow for local topological changes during training.

---

[7]The *repli-score* is a real-valued proxy for the cell cycle, which was introduced in [31], and which can be computed out of the copy numbers of genome regions associated to the early phases of the cell cycle. Thus, a large correlation with this score indicates that the circular shape in the optimized latent space is indeed representative of the cell cycle itself.

## Acknowledgements

M.C. was supported by ANR grant "TopModel", ANR-23-CE23-0014. The authors are grateful to the OPAL infrastructure from Université Côte d'Azur for providing resources and support.

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

## A  A more extensive presentation of TDA

The starting point of TDA is to extract *quantitative* topological information from structured objects—for example graphs, points sampled on a manifold, time series, etc. Doing so relies on a piece of algebraic machinery called *persistent homology* (PH), which informally detects the presence of underlying topological properties in a *multiscale* way. Here, estimating topological properties should be understood as inferring the number of connected components (topology of dimension 0), the presence of loops (topology of dimension 1), of cavities (dimension 2), and so on in higher dimensional settings.

**Simplicial filtrations.**  Given a finite simplicial complex[8] $K$, a *filtration* over $K$ is a map $t \mapsto K_t \subseteq K$ that is non-decreasing (for the inclusion). For each $\sigma \in K$, one can record the value $t(\sigma) := \inf\{t \,|\, \sigma \in K_t\}$ at which the simplex $\sigma$ is inserted in the filtration. From a topological perspective, the insertion of $\sigma$ has exactly one of two effects: either it creates a new topological feature in $K_t$ (e.g., the insertion of an edge can create a loop, that is, a one-dimensional topological feature) or it destroys an existing feature of lower dimension (e.g., two independent connected components are now connected by the insertion of an edge). Relying on a matrix reduction algorithm [20, §IV.2], PH identifies, for each topological feature appearing in the filtration, the *critical pair* of simplices $(\sigma_b, \sigma_d)$ that created and destroyed this feature, and as a byproduct the corresponding *birth* and *death* times $t(\sigma_b), t(\sigma_d)$.[9] The collection of intervals $(t(\sigma_b), t(\sigma_d))$ is a (finite) subset of the open half-plane $\{(b, d) \in \mathbb{R}^2 \,|\, b < d\}$, called the *persistence diagram* (PD) of the filtration $(K_t)_t$. The distance of such a point $(t_b, t_d)$ to the diagonal $\{b = d\}$, namely $2^{-\frac{1}{2}}|t(\sigma_b) - t(\sigma_d)|$, is called the *persistence* of the corresponding topological feature, as an indicator of "for how long" could this feature be detected in the filtration $(K_t)_t$.

Note that if $(\sigma_b, \sigma_d)$ is a critical pair for our filtration $(K_t)_t$, it holds that $|\sigma_b| = |\sigma_d|+1$. The quantity $|\sigma_b| - 1$ is the dimension of the corresponding topological feature (e.g., loops, which are created by the insertion of an edge and killed by the insertion of a triangle, are topological features of dimension one). From a computational perspective, deriving the PD of a filtration $(K_t)_t$ is empirically[10] quasilinear with respect to the number of simplices in $K$ (which can still be extremely high in the case of Vietoris-Rips filtration—see below—where $K = 2^V$ with $|V| = n$ being typically quite large).

**The Vietoris-Rips filtration.**  A particular instance of simplicial filtration that will be extensively used in this work is the Vietoris-Rips (VR) one. Given $X = (x_1, \ldots, x_n) \in \mathbb{R}^{n \times d}$ a point cloud of $n$ points in dimension $d$, one consider the simplicial complex $K = 2^X$ and then the filtration $(K_t)_t$ defined by

$$\sigma = \{x_{i_1} \ldots x_{i_p}\} \in K_t \iff \forall j, j' \in \{1, \ldots, p\}, \ \|x_{i_j} - x_{i_{j'}}\| \leq t. \tag{8}$$

The corresponding persistence diagram will be denoted, for the sake of simplicity, by $\mathrm{Dgm}(X)$.

Note that for $t < 0$, $K_t = \emptyset$, when $t \geq \mathrm{diam}(X)$, $K_t = K$, and there is always a point with coordinates $(0, +\infty)$ in $\mathrm{Dgm}(X)$ accounting for the remaining connected component when $t \to \infty$. This is the unique point in $\mathrm{Dgm}(X)$ for which the second coordinate is $+\infty$ (and is often discarded in practice, as it does not play any significant role). Intuitively, $\mathrm{Dgm}(X)$ reflects the topological properties that can be inferred from the *geometry* of $X$; this can be formalized by various results which state, roughly, that if the $x_i$ are i.i.d. samples from a regular measure $\mu$ supported on a submanifold $\mathcal{M} \subset \mathbb{R}^d$, then with high probability the topological properties of $\mathcal{M}$ are reflected in $\mathrm{Dgm}(X)$ (see [13, 9]). From a computational perspective, note that the VR filtration only depends on $X$ through the pairwise distance matrix $(\|x_i - x_j\|)_{1 \leq i,j \leq n}$, and thus the complexity of computing $\mathrm{Dgm}(X)$ depends only linearly in $d$.[11] On the other hand, since $\mathrm{Dgm}(X)$ scales (at least) linearly

---

[8]A simplicial complex is a combinatorial object generalizing graphs and triangulations. Given a finite set of vertices $V = \{v_1, \ldots, v_n\}$, a finite simplicial complex $K$ is a subset of $2^V$ (whose elements are called *simplices*) such that $\sigma \in K \Rightarrow \tau \in K, \ \forall \tau \subseteq \sigma$ (if a simplex is in the complex, its faces must be in it as well).

[9]It may happen that a topological feature appears at some time $t_b$ and is never destroyed, in which case the death time is set to $+\infty$. However, in the context of the Vietoris-Rips filtration, extensively studied in this work, this (almost) never happens. See the next paragraph.

[10]The theoretical worst case yields a cubic complexity, but the matrix that has to be reduced is typically very sparse, enabling this practical speed up.

[11]However, the statistical efficiency of $\mathrm{Dgm}(X)$ when it is used as an estimator for the topology of an underlying manifold $\mathcal{M}$ deteriorate when the *intrinsic* dimension of $\mathcal{M}$ increases.

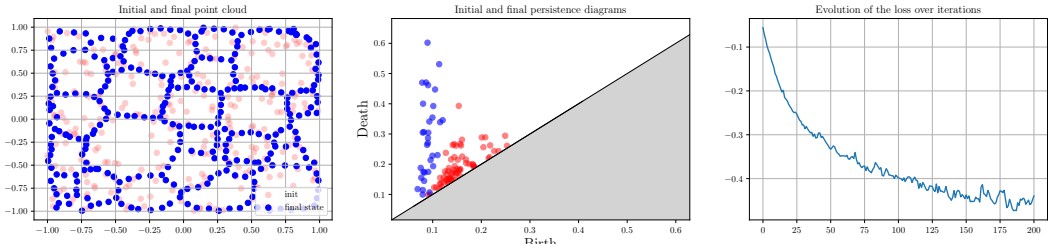

Figure 8: Topological optimization of an initial point cloud $X$ (in red) by minimizing $X \mapsto \sum_{(b,d)\in \mathrm{Dgm}(X)} -|d|^2 + \sum_{x\in X} \mathrm{dist}(x, [-1,1]^2)$. This loss favors the apparition of topological features (loops) while the regularization term penalizes points that would go to infinity otherwise.—Experiment reproduced following the setting of [4], using code available at `https://github.com/GUDHI/TDA-tutorial/blob/master/Tuto-GUDHI-optimization.ipynb`.

with respect to the number of simplices in $K$, computing the whole VR diagram of a point cloud $X \in \mathbb{R}^{n\times d}$ can take up to $O(2^n)$ operations. Even if one restricts topological features of dimension $d' \le d$ (e.g. $d' = 1$ if one only considers loops)—as commonly done—the complexity is of order $O(n^{d'+2})$, which becomes quickly intractable if $n$ is large, even if $d' = 1$ or $2$.

## B   Delayed proofs

*Proof of Proposition 3.2.* One has: $\|\tilde{v}(x) - \tilde{v}(y)\|_1 = \|\sum_{i\in I}(K(x, x_i) - K(y, x_i))(-\mathbf{K}^{-1}\nabla L(\tilde{X}(t)))_i\|_1 \le \sum_{i\in I} |\rho_\sigma(\|x-x_i\|) - \rho_\sigma(\|y-x_i\|)| \cdot \|(-\mathbf{K}^{-1}\nabla L(\tilde{X}(t)))_i\|_1$, since we are using Gaussian kernels. As $|\rho_\sigma(\|x - x_i\|) - \rho_\sigma(\|y - x_i\|)| \le C_{d,\sigma}\|x - y\|_2$, with $C_{d,\sigma} = 2^{\frac{d+1}{2}}\pi^{\frac{d-1}{2}}\sigma^{d-1}$ (see [2, Theorem 8]), it follows that $\|\tilde{v}(x) - \tilde{v}(y)\|_1 \le C_{d,\sigma}\|x - y\|_2 \cdot \|\mathbf{K}^{-1}\|_1 \cdot \|\nabla L(\tilde{X}(t))\|_1$.

Let us upper bound the term $\|\nabla L(\tilde{X}(t))\|_1$. Let us start with the simplification loss, one has $\frac{\partial \ell}{\partial b_i} = 2(b_i - d_i)$ and, writing $b_i = \|x_{i_1} - x_{i_2}\|_2$ (for some critical points $x_{i_1}, x_{i_2} \in \tilde{X}(t)$), one has $\frac{\partial b_i}{\partial x_{i_1}} = \frac{x_{i_1}-x_{i_2}}{\|x_{i_1}-x_{i_2}\|_2}$ and $\frac{\partial b_i}{\partial x_{i_2}} = -\frac{x_{i_1}-x_{i_2}}{\|x_{i_1}-x_{i_2}\|_2}$. Similarly, one has $\frac{\partial \ell}{\partial d_i} = -2(b_i - d_i)$, and writing $d_i = \|x_{i_3} - x_{i_4}\|_2$ provides the corresponding partial derivatives.

Applying (2), this gives:

$$\|\nabla L(\tilde{X}(t))\|_1 = \sum_{x\in \tilde{X}(t)} \| \sum_{x\overset{b,1}{\to}(b_i,d_i)} 2(b_i - d_i)\frac{x - x_{i_2}}{\|x - x_{i_2}\|_2} - \sum_{x\overset{b,2}{\to}(b_i,d_i)} 2(b_i - d_i)\frac{x_{i_1} - x}{\|x_{i_1} - x\|_2}$$

$$- \sum_{x\overset{d,1}{\to}(b_i,d_i)} 2(b_i - d_i)\frac{x - x_{i_4}}{\|x - x_{i_4}\|_2} + \sum_{x\overset{d,2}{\to}(b_i,d_i)} 2(b_i - d_i)\frac{x_{i_3} - x}{\|x_{i_3} - x\|_2}\|_1,$$

where $\overset{b,1}{\to}$ (resp. $\overset{b,2}{\to}$) means that $x$ appears as left point (resp. right point) in the computation of the birth filtration value $b_i$ of one of the $k$ PD points $(b_i, d_i) \in \mathrm{Dgm}(\tilde{X}(t))$ associated to the loss, and similarly for death filtration values. A brutal majoration finally gives $\|\nabla L(\tilde{X}(t))\|_1 \le 2\sqrt{d}\sum_{x\in \tilde{X}(t)} \sum_{x\to(b_i,d_i)} |b_i - d_i| \le 8\sqrt{d} \cdot \mathrm{Pers}_k(\mathrm{Dgm}(\tilde{X}(t)))$, as there are at most four points associated to every $(b_i, d_i) \in \mathrm{Dgm}(\tilde{X}(t))$. One can easily see that the same bound applies to the augmentation loss.

Let us finally bound $\|\mathbf{K}^{-1}\|_1$, one has $\|\mathbf{K}^{-1}\|_1 = \kappa(\mathbf{K})/\|\mathbf{K}\|_1 \le \kappa(\mathbf{K})$. Indeed, as we are using Gaussian kernels, $\|\mathbf{K}\|_1 = \max_{1\le i\le n} \sum_{j=1}^n \rho_\sigma(\|x_i - x_j\|_2) \ge 1$. $\qquad \square$

## C   Complementary experimental results and details

**Subsampling and improving over [4].**   We reproduce the experiment of [4, §5], see also Figure 8, but starting from an initial point cloud $X_0$ of size $n = 2,000$ instead of $n = 300$. This makes the

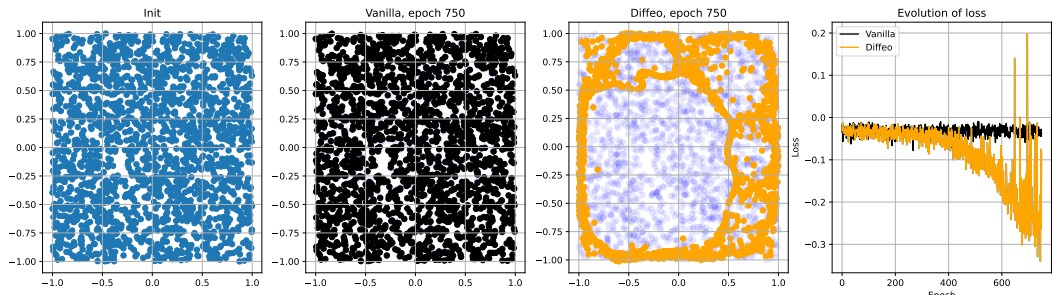

Figure 9: Topological optimization with subsampling. From left to right, the initial point cloud $X_0$, the point cloud after 750 steps of vanilla gradient descent (+subsampling), the point cloud after 750 steps of diffeomorphic interpolation gradient descent (+subsampling), loss evolution over epochs. Parameters: $\lambda = 0.1$, $\sigma = 0.1$.

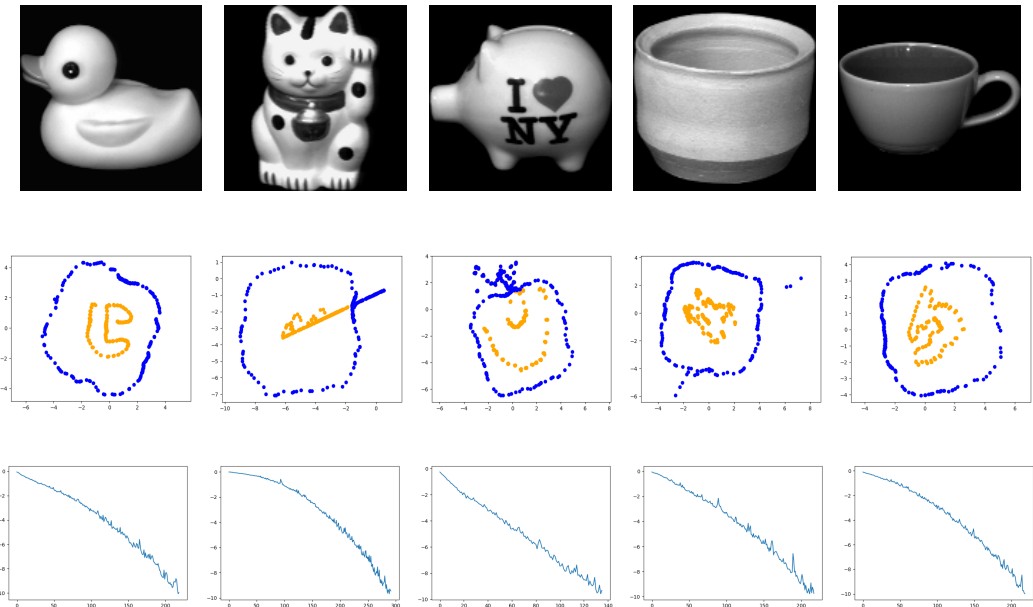

Figure 10: Topologically-optimized LSs and losses for duck, cat, pig, vase and teapot.

raw computation of $\mathrm{Dgm}(X_k)$ *at each gradient* step unpractical. Following Section 3.2 we rely on subsampling with sample size $s = 100$ and apply Algorithm 1. Results are summarized in Figure 9. While relying on vanilla gradients and subsampling barely changes the point cloud even after 750 epochs, the diffeomorphic interpolation gradient with subsampling manages to decrease the loss.

**More extensive reports of running time and comments.**   On the hardware used in our experiments (the first two experiments were run on a `11th Gen Intel(R) Core(TM) i5-1135G7 @ 2.40GHz`, the last one on a `2x Xeon SP Gold 5115 @ 2.40GHz.`), we report the approximate following running times:

- Small point cloud optimization without subsampling (see Figure 4, $n = 200$ points): one gradient descent iteration takes about 1s for the vanilla topological gradient and our diffeomorphic interpolation. The use of `oineus` integrated in our pipeline raises the running time (per iteration) to 10 to 20 seconds. Note that the diffeomorphic interpolation and `oineus` may converge in less steps than the vanilla topological gradient, preserving a competitive advantage. We also believe that `oineus` has a significant room for improvement in terms of running times and may be a promising method in the future to be used jointly with our approach.

- Iterating over the stanford bunny with subsampling ($n = 35,947$, $s = 100$) takes about 3 seconds per iteration for the vanilla topological gradient and 20 second for our diffeomor-

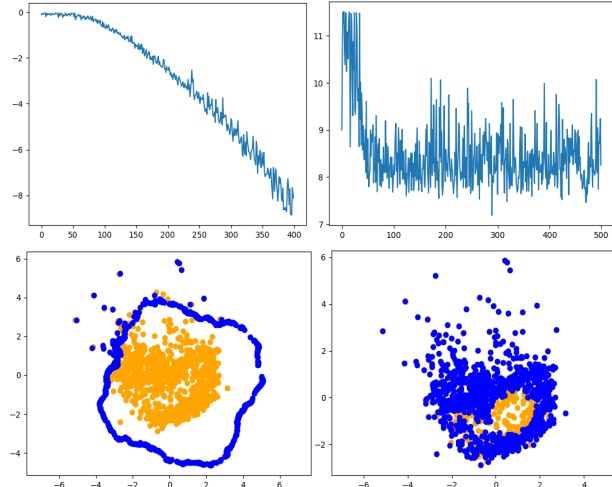

Figure 11: Decrease of augmentation (up left) and registration (up right) losses for the scHiC dataset, as well as the corresponding initial (orange) and optimized (blue) LSs displayed below.

phic interpolation. The increase in running time with respect to the previous experiment mostly lies on instantiating and applying the $n \times d$ ($d = 3$) vector field $\tilde{v}$ (requires to compute $\rho_i(x - x_i)$ for each new $x$ ($n$ of them) and sampled $x_i$ ($|I|$ of them, which is typically very small), hence a $\sim O(n)$ complexity).

- Training the VAE for the COIL and scHiC datasets are the most computationally expensive parts of this work: it takes about 3 hours per image (20 of them), and 3 hours also for the scHiC dataset. In contrast, performing the topological optimization take few dozen of minutes (less than one hour) for each image. Applying it is done in few seconds at most. Recall that our method is designed to handle pre-trained models (which may be way more sophisticated than the one we used!); and its running time does not depend on the complexity of the model.

**Influence of the bandwidth** $\sigma$. We reproduce the same experimental setting as in Figure 3, i.e., sample points uniformly on a circle of radius 1 plus additional noise $\sim \mathcal{N}(0, 0.05I_2)$, and consider minimizing the total persistence of the point cloud. We take $\sigma \in \{0, 0.1, 0.2, 0.3, 0.5, 0.7, 1, 2, 3, 4, 5\}$ (with the convention that $\sigma = 0$ corresponds to the vanilla topological gradient) and learning rate $\lambda = 0.1$. We also rely on a subsampling with batch size $s = 50$. To quantify the variability of the scheme with respect to the randomness induced by the subsampling step, we run each gradient descent 50 times with a fixed initialization $X_0$, up to a maximum of 200 iterations, stopped earlier if a loss of 0 (no topology left, global minimum has been reached) is measured.

Figure 12 displays the results of this illustrative experiment. We report the median of both running time and number of iterations to reach convergence (or reach the 200 iterations limit), along with the 10 and 90 percentiles. The conclusions are:

- For $\sigma = 0$ (vanilla) and $\sigma \geq 3$, the gradient descent never converges in less than 200 steps. Since the radius of the diameter of the circle is 2, it is not surprising that taking a bandwidth larger than that hinders convergence.

- For $\sigma \in (0.1, 1]$, the convergence occurs within the same order of magnitude (between 0.49 and 1.74s), the best performance being reached at $\sigma = 0.3$ (recall that we used in the paper, testifying that we did not rely on hyperparameter tuning). It suggests that, on regular structure, the approach is smooth with respect to $\sigma$. Empirically, we observe that a good proxy is to take $\sigma < \text{median}(\{|x_i - x_j|\}_{i,j})$. Note that even though in theory, $\sigma \to 0$ should recover the vanilla topological gradients, one is limited by numerical accuracy when evaluating the Gaussian kernel.

- The variation around the median over 50 runs is very small: the randomness of the samples at each iteration (hence of the trajectory) barely impacts the decrease of the loss and thus the convergence time.

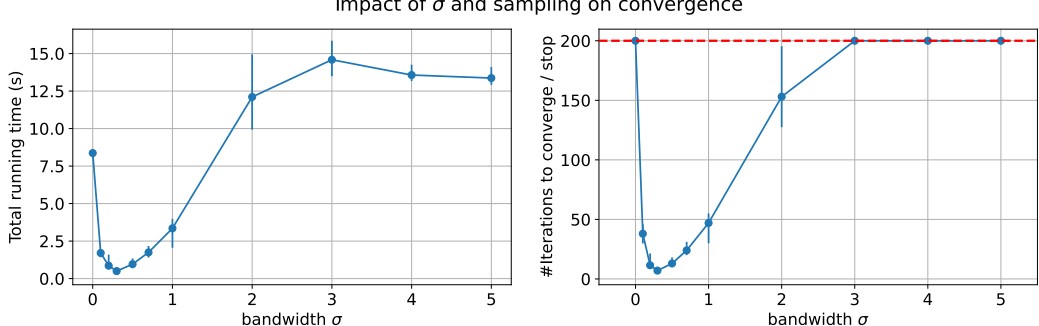

Figure 12: Topological simplification, point cloud of diameter 2 with median pairwise distance $\simeq \sqrt{2}$. Median and 10-90 percentiles over 50 runs. (Left) Time to converge for different values of $\sigma \in [0, 5]$ ($\sigma = 0$ corresponds to Vanilla). (Right) #iterations to converge (or stop after 200 iterations, indicated by the dashed red line).

We also studied how the bandwidth $\sigma$ influences the correlation scores of Table 1 in Figure 13. We observe oscillations for values that are roughly on the sides (very small or very large), and more stable scores for middle range values. Note that these oscillations could also come from how the correlation score itself is computed.

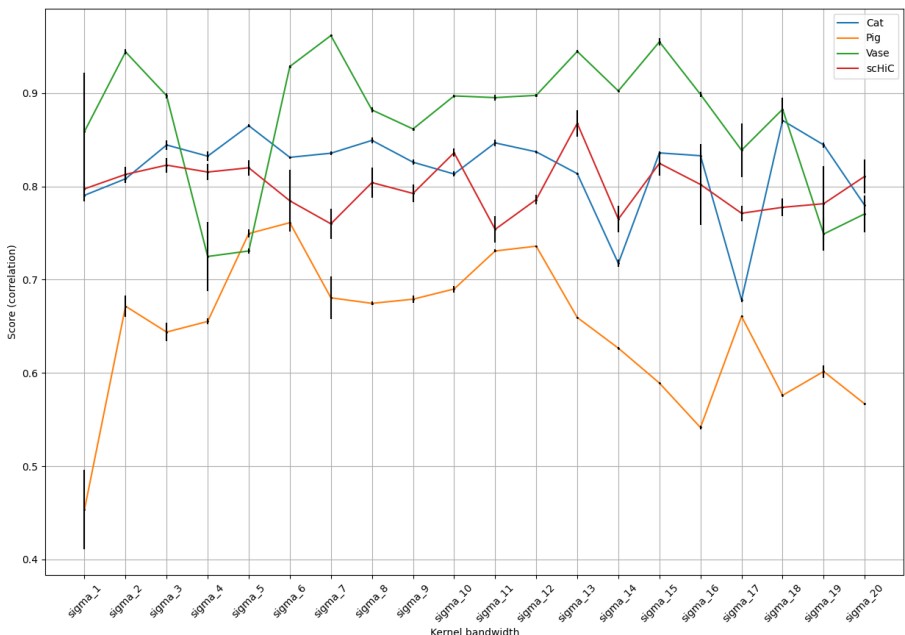

Figure 13: Influence of the kernel bandwidth $\sigma$ on correlation scores for a few datasets. The values of $\sigma$ are evenly spaced between 0.05 and 1 for the COIL datasets, and between 0.025 and 0.5 for the scHiC dataset.

