# OpenReview forum: "Diffeomorphic interpolation for efficient persistence-based topological optimization"
_NeurIPS.cc/2024/Conference — NeurIPS 2024 poster_

### Official Review · Reviewer_uH7P · 2024-06-28

**Soundness:** 3
**Presentation:** 3
**Contribution:** 3
**Rating:** 7
**Confidence:** 4

**Summary:**

The paper introduces a novel approach to topological optimization using diffeomorphic interpolation to tackle the sparsity and inefficiency of gradient descent in topological data analysis (TDA). The authors focus on optimizing point clouds by transforming sparse gradients into smooth vector fields. They achieve this by assigning exponentially decaying weights to each point with a gradient, allowing the influence to span the entire space. This approach enables a larger set of points to be optimized in each iteration, thereby accelerating convergence. The effectiveness of the method is demonstrated through experiments that showcase its fast convergence. Additionally, the method proves to be particularly efficient when combined with subsampling techniques, facilitating large-scale topological optimization.

**Strengths:**

1. In practical applications, optimizing topological loss is often slow. The proposed method significantly accelerates this process.
2. Diffeomorphic interpolation is proven to provide a descent direction for the topological loss, with a provided smoothness bound.
3. The proposed method is especially beneficial for subsampling techniques, leading to both faster convergence and reduced computation time.

**Weaknesses:**

The authors describe three types of topological loss where the proposed method can be applied. However, the application to topological registration losses is not demonstrated in their experiments. My concern is that, in topological registration loss, the match between the output diagram and the target diagram is often unstable. Any mismatch could result in an increased registration loss. With diffeomorphic interpolation, this increment could be even larger, potentially making the overall optimization more unstable.

**Questions:**

The experiments on convergence rate are only designed for simple topological situations. When the topological complexity increases, the optimization becomes unstable. Will the proposed method introduce additional instability to the optimization? In other words, could a convergence theorem for the proposed method be derived?

**Limitations:**

See above.

---

> ### Author Rebuttal · Authors · 2024-08-05
>
> > topological registration losses is not demonstrated
>
> See Global Rebuttal and companion pdf (Figure 2(b) and Table 1), where we showcased our methodology on a topological registration loss on a real-world dataset of single cells. While we do observe oscillations, we still note a global decrease of the loss. As for the correlation score, the improvement is very marginal, which we believe is due to the fact that correlation score only measures the correlation between latent space angles and ground truth. Thus, while the optimized latent space does exhibit a better delineated circular shape with the registration loss (see Figure 2(b)), the size of that circle is prescribed by the coordinates of the target PD point $[-3.5, 3.5]$ (whereas this size is made as large as possible with the augmentation loss), which in turn does not change the latent space angles very much (as well as the correlation score overall) compared to the augmentation loss.
>
> > The experiments on convergence rate are only designed for simple topological situations. When the topological complexity increases, the optimization becomes unstable. Will the proposed method introduce additional instability to the optimization? In other words, could a convergence theorem for the proposed method be derived?
>
> Thank you for the suggestion. Proving a convergence theorem is indeed one of the future directions we aim at pursuing. We believe that such convergence results should be accessible provided that existing results in this vein (such as, e.g, [_Optimizing persistent homology based functions_, Carrière et al., **ICML**, 2021]) can be thought of as particular instances of our setting when $\sigma \to 0$. Hence, convergence should also occur for $\sigma$ and learning rates that are sufficiently small, in order to ensure that they do not interfere too much with the different terms involved by the high topological complexity.
>
> Guaranteeing the convergence for fixed bandwidth $\sigma$ and learning rate $\lambda$ in the greatest generality seems too demanding: we designed in Global Rebuttal and the companion pdf some cases where convergence seems to fail numerically because $\sigma$ is much too large with respect to the diameter of the point cloud. As commented in Global Rebuttal, a good heuristic in our experiments is to take $\sigma$ as a fraction of the median distance in the point cloud.
> Therefore, an appealing generalization suggested by reviewer v7wD would be to use in practice an adaptive / locally varying $\sigma$ that would in some sense, adapt to small vs large topological features. Note that one should do it in a way such that $K(x,y)$ remains positive semi-definite for any $x,y$ though, so that the theory of diffeomorphic interpolations presented in Section 2.2 still applies.

---

> > ### Comment · Reviewer_uH7P · 2024-08-09
> >
> > Thank you for the response. It’s encouraging to hear that global convergence has been observed in experiments. I look forward to seeing a convergence theorem for diffeomorphic interpolation. Overall, this is a strong paper.

---

### Official Review · Reviewer_v7wD · 2024-07-05

**Soundness:** 3
**Presentation:** 3
**Contribution:** 3
**Rating:** 7
**Confidence:** 4

**Summary:**

In "Diffeomorphic interpolation for efficient persistence-based topological optimization", the authors provide a novel method to compute interpolations of gradients of persistence diagrams. This approach makes optimisation of point clouds w.r.t. loss functions defined on the associated persistence diagrams feasible, as it provides generalised gradients which smoothly vary over the underlying space. In summary, the authors use some form of Gaussian kernels and ideas from stochastic gradient descent to extend the standard gradients only defined on a few points to the entire  base space. They compare the running time of their algorithm favourably to other existing topological gradients, and perform qualitative experiments on synthetic data and the latent space of an autoencoder.

**Strengths:**

1. The paper presents a novel combination of topological gradients and diffeomorphic interpolation. The authors do a good job of embedding this approach into the existing literature and highlighting its differences and relevancy.
2. The submission is technically sound and provides theoretical guarantees which are proven. The application to the latent space of variational auto encoders provides a nice and simple application adding to the theoretical contributions.
3. The proposed method significantly enhances the applicability of optimisation of point clouds w.r.t. to topological loss functions. The idea to solve some of the problems of topological gradients using diffeomorphic interpolation is new and very good.
(4. The gif is very nice :) )

**Weaknesses:**

1. There is no intuition provided behind theoretical result number 2. One can either just believe it, or read the very brief and technical derivation in the appendix; a brief summary and intuition in the main text would be great!
2. While there are theoretical guarantees for the derivative of the loss at a single time point (and thus in the step-size -> 0 case), any guarantees which also apply in the case for discrete steps are missing. To the reviewer's understanding, the diffeomorphic interpolation of the loss function does not need to be continuous in t, even in the case of infinitesimal small step sizes. Thus the relevancy of the theoretical guarantees in practice is not clear.
3. The choice of the bandwidth of the Gaussian kernel used for the diffeomorphic interpolation, a seemingly very important hyperparameter, is not discussed in the paper. The impact of different choices of sigma is not looked at in the experiments. (See the question section.)
4. The experiments presented are either toy examples or very artificial. An application on a down-stream task relevant in some field of practice or real-world application would significantly enhance the strength of the paper. (This is a caveat applicable to much of the TDA literature in general though.)

**Questions:**

1. What is the intuition behind the proof of Theorem 2? (Cf. Weakness 1)
2. What do the theoretical guarantees tell us in practice? Do they tell us something? Can you provide any guarantees on the evolution of the loss over a number of steps with a discrete step size? Or even in the continuous case? If not, what are the theoretical reasons these cases are so much harder?
3. How did you choose the bandwidth in practice? If someone knows nothing about their dataset, what would be a good heuristic to choose sigma? Do we need to fix a global sigma or can we vary sigma locally?
4. What is the impact of different choices of sigma on the results of the experiments? How sensitive are the experiments to varying sigma?
5. In the bottom part of Figure 4, why is the loss of Oineus significantly larger than the loss of the other methods for small x values?
6. In Figure 3, how did you pick the hyperparameters? Did you tune them for the diffeomorphic interpolation, or for the vanilla case? It might be the case that different hyperparameter regimes are optimal.
7. In Figure 9, 3rd from the left: what are the purple points? 4th from the left: How can the large loss spikes for the diffeo algorithm be explained? How do the hyperparameter choices influence the loss stability?

I would like to thank the authors in advance for their answers!

Comments:
* Line 20: "can be done once and then re-applied to new data in linear time" -> What does that even mean?
* ll 33: "active line OR research"
* ll 37: "yielding to the problem"
* ll 84: "allows to process datasets whose sizes are currently out of reach in TDA," This claim is a bit bold: Subsampling and taking landmarks was a thing even before this paper. Maybe this applies to optimisation of point clouds in TDA:
* ll 100: "machinary"
*  section 2.2: The defintion of RKHS is really confusing. If I didn't know the definition beforehand, I would not have understood it there
* ll 181: where does v live? can X and Y be any arbitrary point clouds? With which definitions? How can X be the argument on the left side of the definition, and still be referenced as all X in the set builder notation on the RHS?
* ll 193: "i-th line"
* ll 214: "s-points"
* ll 476: The grammar of this sentence is confusing.
* page 13, footnote 5: only non-empty simplices. footnote 8: "deteriorate"

**Limitations:**

As detailed above, a discussion on hyperparameter choices, the stability of the algorithm, and the relevance of the guarantees in practice is missing. For possible negative societal impact: In Figure 5 the authors blow up a synthetic bunny and make the forgivable mistake to fail to mention that this experiment should not be performed on real-world bunnies...

---

> ### Author Rebuttal · Authors · 2024-08-05
>
> > 1. What is the intuition behind the proof of Theorem 2?
>
> Our goal with this result was to quantify the smoothness of our proposed diffeomorphic interpolation (as characterized with its Lipschitz constant) based on the kernel it is computed from. In the case of the Gaussian kernel, we found that the Lipschitz constant depends on the kernel bandwidth $\sigma$: indeed, the more spread the Gaussian is, the more critical points (identified during the computation of PH) can influence other data points potentially far from them, introducing unwanted distortions and larger Lipschitz constant. Similarly, if the condition number is large, inverting the kernel matrix might introduce instabilities in the $\alpha_i$ in Equation (4), and thus a larger Lipschitz constant as well. Finally, the total persistence also appears, as the more PD points one has to optimize, the more critical pairs will appear, and thus the more constrained the minimization problem in section 2.2 is, leading to more complex diffeomorphic interpolation solutions with larger Lipschitz constants.
>
> > 2. What do the theoretical guarantees tell us in practice? (…) evolution of loss with discrete step-size? (…) what makes it much harder?
>
> We agree that the theoretical guarantees remain of limited use in practice: they only tell us that our diffeomorphic interpolations also provide meaningful descent directions for the loss; and thus that for a sufficiently small step-size, the loss must decrease. Empirically, we showcase that overall, the loss decreases substantially faster using our method. Proving this formally in some reasonable settings (continuous-time, etc.) remains quite hard though, for various reasons:
> - the losses considered are typically not convex (with respect to the input point cloud $X$), and thus it is hard to quantify how large the learning rate can be taken and how much the loss decreases after a discrete-time step,
> - Our $\tilde{v}$ that dictates the dynamic of the flow is not a gradient (at least for Euclidean geometry), so we cannot apply faithfully the gradient flow/descent literature to get theoretical guarantees,
> - The dependence of the Vietoris--Rips filtration on the point cloud $X$ is cumbersome. We can derive theoretical guarantees as long as the ordering of the pairwise distances $\|x_i - x_j\|$ remains unchanged (i.e., as long as the points are moved by less than $\min_{ij,kl} |\|x_i - x_j\| - \|x_k - x_l\| |$), but this is fairly restrictive.
>
> It is clear that studying this model is of interest, but we believe that it may be postponed for future work.
>
> > 3. and 4. How did you choose the bandwidth in practice? (…) heuristic ? (…) vary sigma locally ? (…) different choices of sigma ?
>
> We chose $\sigma$ with no parameter tuning (see Global Rebuttal and companion pdf, which actually show that the results of our method in Figure 3 can actually be improved if we pick $\sigma = 0.3$ instead of $0.1$); the heuristic being that $\sigma$ should be a fraction of the median distance between input points (this heuristic is common in calibration of Gaussian kernel in other statistical scenarios).
>
> It may be possible to consider a space-dependent $\sigma$ or other variant: as long as $(x,y) \mapsto K(x,y)$ remains valued in positive semidefinite matrices, the theory presented in Section 2.2. adapts faithfully (see also line 154 in the main paper).
>
> We included in Global Rebuttal and companion pdf (Figures 1 and 2(a)) some experiments for varying $\sigma$, showcasing the robustness of our approach w.r.t. $\sigma$.
>
> > 5. Oineus loss larger
>
> Good catch, thanks! We forgot a normalization factor in the way the oineus loss is displayed. Fortunately, as our stopping criterion is that a loss of exactly $0$ should be reached (see line 248 in the main paper), this does not change the number of iterations (e.g., 49 iterations for oineus alone) nor the running times, and thus the conclusions of this proof-of-concept experiment remain unchanged. We will correct this in the next version.
>
> > 6. In Figure 3, how did you pick the hyperparameters? (…) It might be the case that different hyperparameter regimes are optimal.
>
> We actually pick $\lambda = 0.1$ for every experiments, with no specific tuning. Given that our approach is directly extrapolating the vanilla topological gradient, we believe that picking the same learning rate for both models is the natural choice.
>
> > Figure 9, (a) 3rd from left (…) purple points? (b) 4th (…) loss spikes for the diffeo algorithm be explained?
>
> (a) For 3rd Figure, transparent blue points simply represent the initial point cloud as a reference. Orange points represent the final state of the point cloud.
>
> (b) We report the loss on the subsample batch (because evaluating the loss on the whole point cloud is computationally prohibitive, see line 227). Because of the stochastic nature of our approach, it may happen that this estimated loss is, on a few occasions, quite high because the subsample does not reflect the expected topological structure.
>
> This does not happen with vanilla gradient descent for a simple reason: the initial point cloud is broadly uniform, so are the subsamples (with high probability) yielding a loss around $-0.03$, and because _vanilla gradient descent fails to perform any significant update of the point cloud_, this situation remains through iterations.
> In contrast, because our diffeomorphic interpolations do change the point cloud, they reach configurations where some subsamples may yield a high loss, though the loss decreases overall.
>
> In a nutshell, these spikes are not related to the optimization technique involved (if the vanilla gradient descent was initialized with, say, the epoch 600 reached by our approach based on diffeomorphisms, one would observe similar spikes), but to the “topological variance” in the subsample for a given configuration, the point being that vanilla gradient descent does not reach such configurations.

---

> > ### Comment · Reviewer_v7wD · 2024-08-12
> >
> > Thank you very much for your detailed rebuttal! I believe your changes and proposed changes to the manuscript are very valuable. I still have some brief comments:
> >
> > 1. I believe that adding the explanations and intuition you gave me in the rebuttal would be a valuable addition to the (appendix of the) paper. For example, sometimes it is very relevant and interesting information why a theorem takes the form it has, and why it is hard to prove something more difficult.
> > 2. I appreciate your study on the effect of the bandwidth $\sigma$ on the performance of your algorithm. I also value the heuristic for choosing $\sigma$. However, to fully appreciate this I think you should use __(a)__ the $\sigma$ computed by this heuristic in your experiments or __(b)__ at least report the suggested $\sigma$ in comparison to the $\sigma$ used in your experiments.
> > 3. I am not convinced by the additional experiments on the VAE. (Or I might have misunderstood.) I know that you have added a loss-term to optimise for the embedding of the VAE having a circular structure. However, did you check whether this particular circle, and the individual location of the points on the circle correspond to any meaningful real-world features? I know that, because of the setup of the experiments and the studied cell-cycle, it is not unreasonable to expect some circle to form. I just don't understand how to quantitatively evaluate the meaningfulness of your result. (Maybe using circular coordinates, for example from [1])
> > 4. I know that this is a complicated question: What would you need to do to generalise your topological optimisation to work with DTM filtrations? It seems that many topological features in noisy real-world settings are much more likely to be uncovered by DTM filtrations, for example in the scihc embedding. [2]
> >
> > Despite all the above remarks, I believe that this is a very cool paper even in its current state. I am looking forward to exciting applications of this in the world of ML. I will thus be raising my score and recommending the paper for acceptance
> >
> > [1] Perea, Jose A. "Sparse circular coordinates via principal ℤ-bundles." Topological Data Analysis: The Abel Symposium 2018. Cham: Springer International Publishing, 2020.
> > [2] Anai, H., Chazal, F., Glisse, M., Ike, Y., Inakoshi, H., Tinarrage, R., & Umeda, Y. (2020). DTM-based filtrations. In Topological Data Analysis: The Abel Symposium 2018 (pp. 33-66). Springer International Publishing.

---

> > > ### Author Response · Authors · 2024-08-12
> > >
> > > _I believe that adding the explanations and intuition you gave me in the rebuttal would be a valuable addition to the (appendix of the) paper (...)_
> > >
> > > _I appreciate your study on the effect of the bandwidth on the performance of your algorithm (...) to fully appreciate this I think you should use (a) the $\sigma$ computed by this heuristic in your experiments or (b) at least report the suggested $\sigma$ in comparison to the $\sigma$ used in your experiments._
> > >
> > > Thank you for your suggestions. We will include both our theorem's discussion and highlight our heuristic and the corresponding $\sigma$ value in the experiment section.
> > >
> > > _I am not convinced by the additional experiments on the VAE (...) did you check whether this particular circle, and the individual location of the points on the circle correspond to any meaningful real-world features?_
> > >
> > > This is a rightful question, and we apologize for not getting into much details about it in the rebuttal (mostly due to lack of space). In order to make sure that the circular shape that we end up with (after topological optimization) is biologically relevant and not artifactual, we measured the correlation between the latent angles (computed with the same formula than the one in the main paper), and a quantity called the **repli-score**, which is computed out of the copy numbers of genome regions associated to the early phases of the cell cycle. It was introduced and provided in [_Cell-cycle dynamics of chromosomal organization at single-cell resolution_. Nagano et al. **Nature**, 2017] as an efficient, computable, and real-valued proxy for the cell cycle. Thus, a large correlation indicates that the circular shape in the optimized latent space is indeed representative of the cell cycle itself.
> > >
> > > _What would you need to do to generalise your topological optimisation to work with DTM filtrations? (...)_
> > >
> > > Thank you for this suggestion. As far as our method is concerned, one only requires a sparse gradient on a point cloud in order to run. Hence, the main question is how to compute such gradients for DTM-based filtrations. For instance, DTM-Rips filtrations should be possible, as the equation for computing the DTM is easy to differentiate, and the gradient for Rips is already known, but DTM-Alpha filtrations should be more difficult, as it is less clear what the gradient of the Alpha filtration is. See also our answer to the last question of reviewer fiZD in the context of multiparameter persistence (which can also be used for dealing with noisy data).

---

> ### Comment · Area_Chair_qWqm · 2024-08-10
>
> Dear reviewer v7wD,
>
> Thank you for your review. The authors have tried to address your concerns in the rebuttal. Please carefully read their rebuttal, and let them know what you think, and whether there is any more clarifications you require. Note that author-reviewer discussion ends on August 13th.
> Thanks!
> the AC

---

### Official Review · Reviewer_GJy4 · 2024-07-12

**Soundness:** 4
**Presentation:** 3
**Contribution:** 4
**Rating:** 7
**Confidence:** 3

**Summary:**

The paper proposes a novel way based on diffeomorphic interpolation to deal with the problem of sparse gradients when performing topological optimisation, which is relevant in the intersection between Topological Data Analysis (TDA) and ML.

**Strengths:**

(S1) The paper addresses a relevant problem in the TDA field

(S2) The paper is comprehensive and well-written

(S3) The paper proposes interesting applications of the proposed method

**Weaknesses:**

(W1) The experimental setup could be strengthen by providing a real-world application

**Questions:**

(Q1) In Figure 7, the initial LS is in blue and the final is in orange, while in Figure 6, the initial LS is in orange and the final is in orange, I find this confusing, is there a reason for it?

(Q2) Can you please elaborate more on the importance of your suggested application to black-box autoencoder models, and how does your method compare to other topological optimisation techniques such as Oineus (as in the table in this section you have only compared your approach to the Vanilla, and you reported Oineus for the rest of the experiments)

(Q3) The performance between Vanilla and Diffeo is very similar in the table in section Black-box autoencoder models (apart from Pig), have you ran your experiments multiple times and what is the mean and standard deviation? Also, do you have an explanation as to why your method outperforms Vanilla by a large margin for Pig?

(Q4) Figure 3 is difficult to interpret. What is considered to be the expected end result of topological optimisation? The final shape after 100 iterations of your algorithm is very distorted, can you please add more details on how is that good?

(Q5) What are some real-world uses of topological optimisation? Can you please elaborate on when one should care about this, and what possible applications are there

(Q6) Do you see application in the real-world of your method, and how big datasets can your approach handle?

(Q7) Can you please elaborate on how exactly your method offers better interpretability for the black-box autoencoder application?

(Q8) How robust is your method to taking different samples and to different sampling strategies?

**Limitations:**

Yes

---

> ### Author Rebuttal · Authors · 2024-08-05
>
> > (Q1) color swap in Figures 6 and 7
>
> This was accidental. Thank you for catching this error!
>
> > (Q2) elaborate on (…) black-box AE models, (…) compare to other topological optimisation techniques
>
> Sorry: we made a misleading mistake by accidentally naming “Vanilla” the competitor of “Diffeo” (ours) in the Table in page 9. Here, “Vanilla” is supposed to mean _no topological optimization on the latent space_ (like “Vanilla VAE”, not “Vanilla topological gradient descent”). We will change this in the revised version and are deeply sorry for the confusion it yielded.
>
> We do not compare our method with existing topological optimization methods (Oineus, Vanilla, etc.) in this experiment. That is for a good reason: the experiment we propose can only be made using our method. Indeed, given a pre-trained VAE $D \circ E$, we design $\varphi : \mathbb{R}^{d’} \to \mathbb{R}^{d’}$ such that $\varphi \circ E$ induces a “better” latent representation (in terms of topological properties) by concatenating our diffeomorphic interpolations $\tilde{v}_k, k=1,\dots,T$.
>
> In contrast, **existing methods do not provide maps defined on the latent space**. Their gradients are only defined on training observations. Given (say) a new latent representation $z$, there is no way to “correct it” using vanilla or oineus gradients.
>
> Aside from computational efficiency, this is the biggest edge of our method over pre-existing topological optimization techniques: having access to maps defined on the whole space opens the way for new applications.
>
> > (Q3) ran experiment multiple times?
>
> See Global Rebuttal and companion pdf, Table 1. As for the Pig result, the large margin is because the initial space (from the VAE) was particularly bad (topologically), see Figure 2(b).
>
> > (Q4) Figure 3 (…) expected end result of topological optimisation? The final shape (…) is very distorted, (…) how is that good?
>
> The loss that we minimize in this PoC experiment is proportional to the topological information contained in the input point cloud. That is, we want to destroy the topological structure (here, the presence of a loop). A good output (low loss) is a point cloud with no underlying loop. The take-home messages of this experiment are:
> - The vanilla gradient only updates a few points (sparse gradient) of the (small) subsample. This yields a low decrease in the loss: after $t=100$ steps, the point cloud has barely changed.
> - In sharp contrast, our diffeomorphic interpolation updates _the whole point cloud_ at each step. It yields a much faster optimization (showcased also in Figure 4), and the whole point cloud is deformed smoothly from the input one.
> - In terms of raw topological loss, there is no “better” final output: both methods manage to eventually provide a point cloud with trivial topology. Nonetheless, having a smooth, invertible, deformation between the input and final objects can be useful and more interpretable; this is for instance leveraged in the black-box AE experiment.
>
> > (W1, real-world experiment) + (Q5) real-world uses of topological optimisation?
>
> See Global Rebuttal and the companion pdf (Figure 2(b) and Table 1) for a real-world application on single-cell data. In a nutshell, topological optimization can be useful if you have a topological prior (e.g., "my model should exhibit a periodic / cyclic structure"). One can also consider the references given in lines 40--42 in the main paper for other possible applications of topological optimization.
>
> > (Q6) how big datasets can your approach handle?
>
> Diffeomorphic interpolation updates computed from a subsample of the dataset modify the whole input point cloud, a sharp difference with the vanilla gradients that only update a fraction of the original point cloud at each step. Our PoC experiments illustrate this (in particular Figure 5 shows that our approach can seamlessly perform topological optimization on a point cloud with $\sim 36,000$ points, a scale completely out of reach of previous methods), and we confirmed this on our new real-world application, which involves Vietoris-Rips PH on $1,171$ single cells. While this is already hardly tractable for vanilla gradients, subsampling combined with our diffeomorphic interpolations allows to perform the task fastly and efficiently.
>
> In terms of computational complexity, the overhead induced by our method compared to vanilla gradients are:
> - inverting the kernel matrix $\mathbf{K}\in \mathbb{R}^{|I| \times |I|}$ where $I$ is the set of indices for which the vanilla gradient is non-zero. Since the vanilla gradient is usually sparse, $I$ is small and this matrix inversion is basically negligible.
> - Computing $\tilde{v}$ on the whole input point cloud $X$, which scales _linearly in the number of points_.
>
> > (Q7) elaborate on (…) interpretability for black-box autoencoder?
>
> The idea showcased in this experiment is the following: each dataset is a collection of images representing a rotating object, hence it is natural to assume that there must be a 2-dimensional underlying circle (topological prior).
> When training a VAE on such a dataset, we obtain an encoder $E$ and a decoder $D$ that enables the generation of new data. However, the latent space does not exhibit the expected circle structure (it has no reason to do so).
> We change the latent space representation using the diffeomorphism $\varphi$ optimized so that $(\varphi \circ E(x_i), i=1,\dots,n )$ satisfies our topological prior ($(x_i)_{i}$ is our training set). Since $\varphi$ is invertible, we can generate new observations using $D \circ \varphi^{-1}$.
>
> This increases interpretability: after topological correction, the latent representation is organized on a circle, and angles in the latent space reflect angles between rotated images. You can now sample "an image rotated by 30°” (wrt some reference image) by simply taking an angle of 30° in the latent space.
>
> > (Q8) How robust (…) to (…) different samples
>
> See Global Rebuttal and companion pdf, Figure 1.

---

> > ### Comment · Reviewer_GJy4 · 2024-08-12
> >
> > I thank the authors for their efforts and the elaborative response. I have read the response and would like to stick with my score.

---

### Official Review · Reviewer_fiZD · 2024-07-12

**Soundness:** 3
**Presentation:** 3
**Contribution:** 3
**Rating:** 7
**Confidence:** 4

**Summary:**

This paper proposes a new topological optimization scheme mainly due to the sparsity of topological gradients. The authors introduce the notion of diffeomorphic interpolation and use this to create a smooth vector field over the whole space, which gives a gradient descent algorithm. The authors show some theoretical results which guarantee that the smoothness of the interpolation can be controlled. Further, the authors show numerical experimental evidence for the quick convergence of their method. The authors show, using experiments, that this scheme can be used to interpolate gradient on the whole space by considering a gradient on a small sample yielding expected results. The authors show that this framework can be used to infuse topological information in a pre-trained autoencoder model.

**Strengths:**

1. Overall, the paper is well-written and organized.

2. The proposed method converges faster than its competitors, as shown in Figure 4.

3. The idea of diffeomorphically interpolating the gradient on a subsample to the entire sample, to the best of my knowledge, is novel and quite useful in practice.

4. The concept of inducing topological information into the latent space of a pre-trained autoencoder is also new to the best of my knowledge.

**Weaknesses:**

1. Section 3 is notation intensive. For example, Proposition 3.2, concepts like condition number are not defined earlier and hence, the expression can be moved to the appendix with the main text containing a line which says that the Lipschitz constant can be bounded which guarantees the smoothness.

2. Proof of Proposition 3.1 can be moved to the appendix as it is not hard to see, given the way \tilde v is defined/constructed to improve the readability.

3. For the subsampling experiment, the comparison with vanilla topological gradient seems slightly unfair because there is no way for the network to update gradient over the whole space when a small percentage of points are being sampled for gradient update. Updating gradient over the whole space is not feasible, but is there a threshold on the number of points until which the vanilla gradient update is feasible and if the final output, in that case, is comparable to the output from diffeomorphic gradient descent?

Minor:

In Algorithm 1, should it be “Set $X_k \coloneqq X_{k-1} - \lambda \tilde v(X_{k-1})$"?

**Questions:**

Can this method work for differentiable multiparameter vectorization methods?

**Limitations:**

Yes, the authors have discussed limitations.

---

> ### Author Rebuttal · Authors · 2024-08-05
>
> > Weaknesses 1. and 2. (Section 3 is notation intensive. // Proof of Proposition 3.1 can be moved to the appendix)
>
> Thank you for the suggestion. We will alleviate Section 3, stating Proposition 3.2. more informally and deferring the complete, technical statement to the appendix. As far as the proof of Proposition 3.1 is concerned, as it is very short, we consider keeping it in the main paper as it shows in a simple way how $\tilde v$ interplays with the loss, which is helpful for developing intuition.
>
> > Weakness 3. (For the subsampling experiment, the comparison with vanilla topological gradient seems slightly unfair because there is no way for the network to update gradient over the whole space)
>
> We agree. One could say that it is precisely the purpose of this experiment: since the vanilla topological gradients are not defined on the whole space (not even the whole input point cloud), and because of the inherent computational limitation of (Vietoris-Rips) persistent homology, it is expected that vanilla gradient descent will perform poorly on this task.
>
> So yes, to some extent, the comparison is indeed unfair, but this is exactly the point we wanted to emphasize: extending gradients via diffeomorphic interpolations has a crucial impact in terms of computational efficiency, opening the way for new experiments involving topological optimization that are not currently accessible with vanilla gradient descent.
>
> > Minor (Should $X_t$ be $X_k$ ?)
>
> Indeed, that’s a typo. Thank you for catching it!
>
> > Question : Can this method work for differentiable multiparameter vectorization methods?
>
> Our approach may be generalized easily: as long as one is given a (sparse) gradient on a point cloud $X \in \mathbb{R}^{n \times d}$, one can build its diffeomorphic interpolation and use the resulting $\tilde{v}$ to update $X$ instead.
>
> Therefore, any methods that returns such gradients can benefit from our approach. This includes recent pipelines involving differentiable multi-parameter PH such as D-GRIL [_D-GRIL: End-to-end topological learning with 2-parameter persistence._ Mukherjee et al. arXiv:2406.07100, 2024] and differentiable signed measures [_Differentiability and convergence of filtration learning with multiparameter persistence._ Scoccola et al. **ICML**, 2024]. In both methods, gradients can be obtained by computing free resolutions of multi-parameter persistence modules, and are thus also very sparse (similar to the 1-parameter case studied in our work) as they depend only on the multigraded Betti numbers of the module.

---

> > ### Comment · Reviewer_fiZD · 2024-08-08
> >
> > I thank the authors for their efforts and the response. I have read the response and would like to stick with my score.

---

### Author Rebuttal · Authors · 2024-08-05

# Global rebuttal

We thank the reviewers for their constructive reviews and overall positive comments on our work. We will take into account all comments on clarity, typo, etc., that have been reported by the reviewers in the revised version of our work.
In addition to individual responses, we provide complementary experimental results hopefully addressing remaining concerns in our attached companion pdf:

- 1. A proof-of-concept study showcasing the robustness of our method to (i) the choice of $\sigma$ as suggested by Reviewer v7wD and (ii) the randomness induced by the subsampling as suggested by Reviewer GJy4 (Figure 1). The influence of $\sigma$ over the correlation scores is also provided (Figure 2(a)).
- 2. A real-world experiment (suggested by Reviewers GJy4, v7wD) on single-cell genomics that involves minimizing a _topological registration loss_ as suggested by Reviewer uH7P (Figure 2(b)).
- 3. The results of the black-box VAE experiments averaged over 100 runs as suggested by Reviewer GJy4 (Table 1), in which we compute the mean scores and standard deviations after adding for each run random uniform noise (with amplitude $0.05$ for COIL and $0.01$ for the single-cell dataset).

We describe below the experiments we ran (for 1. and 2.; 3. is exactly the one in the paper but averaged over 100 runs).

**1. Influence of $\sigma$ and subsampling (PoC).** We reproduce the same experimental setting as in Figure 3 of the main paper, i.e. sample points uniformly on a circle of radius $1$ plus additional noise $\sim \mathcal{N}(0,0.05 I_2)$, and consider minimizing the total persistence of the point cloud.
We take $\sigma \in \{ 0, 0.1, 0.2, 0.3, 0.5, 0.7, 1, 2, 3, 4, 5 \}$ (with the convention that $\sigma = 0$ corresponds to the vanilla topological gradients and learning rate $\lambda = 0.1$ (as in the paper).
We rely on a subsampling with batch size $s=50$ (as in the paper).
To quantify the variability of the scheme (suggestion by reviewer GJy4) with respect to the randomness induced by the subsampling step, we run each gradient descent $50$ times with a fixed initialization $X_0$, up to a maximum of $200$ iterations, stopped earlier if a loss of $0$ (no topology left, global minimum has been reached) is measured.

Figure 1 in the companion pdf displays the results of this illustrative experiment.
We report the median of both running time and number of iterations to reach convergence (or reach the $200$ iterations limit), along with the $10\%$ and $90\%$ percentiles.
The conclusions are:
- For $\sigma = 0$ (vanilla) and $\sigma \geq 3$, the gradient descent _never converges_ in less than 200 steps. Since the radius of the diameter of the circle is $2$, it is not surprising that taking a bandwidth larger than that hinders convergence.
- For $\sigma \in (0.1,1]$, the convergence occurs within the same order of magnitude (between $0.49$ and $1.74$s), the best performance being reached at $\sigma = 0.3$ (recall that we used $\sigma = 0.1$ in the paper, testifying that we did not rely on hyperparameter tuning). It suggests that, on regular structure, the approach is smooth with respect to $\sigma$. Empirically, we observe that a good proxy is to take $\sigma < \mathrm{med}((\|x_i - x_j\|)_{ij})$. Note that even though it theory, $\sigma \to 0$ should recover the vanilla topological gradients, one is limited by numerical accuracy when evaluating the Gaussian kernel.
- The variation around the median over 50 runs is very small: the randomness of the samples at each iteration (hence of the trajectory) barely impacts the decrease of the loss and thus the convergence time.

As for correlation scores in Figure 2(a), we observe oscillations for $\sigma$ values that are roughly on the sides (very small or very large), and more stable scores for middle range values. Note that these oscillations could also come from how the correlation score itself is computed.

**2. Real-world application on single-cell data.** We also designed an experiment on single cell HiC data inspired from [_A Gradient Sampling Algorithm for Stratified Maps with Applications to Topological Data Analysis_. Leygonie et al. **Math. Prog.**, 2023]. The dataset is comprised of single cells characterized by chromatin folding, that is, each cell is encoded by the distance matrix of its DNA fragments. The dataset we focus on is taken from [_Cell-cycle dynamics of chromosomal organization at single-cell resolution_. Nagano et al. **Nature**, 2017], in which it was shown that cells are sampled at different phases of the cell cycle. Thus, we expect latent embeddings of this dataset to exhibit a circular shape, that we can constrain with diffeomorphic topological optimization.

Specifically, we processed this single cell dataset of $1,171$ cells with the stratum-adjusted correlation coefficient (SCC) with $500$kb and convolution parameter $h=1$ on chromosome $10$. Then, we run kernel PCA on the SCC matrix to obtain a preprocessed dataset of dimension $100$, on which we applied the same VAE architecture than the one used for COIL data in the main paper. Finally, we optimized the same augmentation loss than for the COIL data, as well as the following registration loss:

$$L :  X\mapsto W_{2}^2({\rm Dgm}^1(X), {\rm Dgm}^{1, t}),$$

where ${\rm Dgm}^1(X)$ contains the PD points of $X$ with distance-to-diagonal at least $\tau=1$, ${\rm Dgm}^{1,t}$ is a target PD with only one point $p^*=[-3.5,3.5]$, and $W_{2}$ is the 2-Wasserstein distance. We used $\sigma=0.25$ (aug.) and $\sigma=0.025$ (reg., sigma is smaller to mitigate the effects of matching instability), $\lambda=0.1$ on $500$ epochs, with subsampling of size $s=300$ for computational efficiency (as computing Vietoris-Rips PH without radius thresholding on $1,171$ points already takes few minutes on a laptop CPU, which becomes hardly tractable if done repetitively as in gradient descent), and we measured correlation between latent space angles and repli scores in Table 1.

---

### Decision · Program_Chairs · 2024-09-25

**Decision:**

Accept (poster)

**Comment:**

Topological optimization suffers from very sparse gradients which leads to slow convergence. This paper suggests a method to alleviate this difficulty by a smoothening procedure. The reviewers were all convinced by the merit and practical benefits  of the method and recommended acceptance.